# Functional specialisation of multisensory temporal integration in the mouse superior colliculus

Gaia Bianchini[1], Ines Razafindrahaba[1], Marcelo J. Moglie [1], George Konstantinou[2], Xavier Cano-Ferrer [2], Albane Imbert[2] & M. Florencia Iacaruso [1] ✉

Our perception of the world depends on the brain's capacity to integrate information from multiple senses, with timing differences serving as crucial cues for binding or segregating cross-modal signals. The superior colliculus (SC) is a central hub for such integration, yet the contributions of its distinct regions remain poorly understood. Here we show, from recordings of over 5000 neurons in awake mice, that multisensory neurons reliably encode audiovisual delays through nonlinear integration of auditory and visual inputs. This nonlinearity enhances the precision of delay representation, with posterior-medial SC populations representing the peripheral sensory field showing superior temporal discriminability. Connectivity analyses reveal stronger coupling in the medial SC and function-specific recurrent networks, with multisensory neurons receiving about half of their local input from other multisensory neurons. Together, these results demonstrate how nonlinear integration, regional specialisation, and network architecture combine to support robust sensory binding and accurate encoding of temporal multisensory information.

The evolution of neural circuits has shaped their structure and ability to extract information from the physical world, optimising organisms' survivability and reproductive success. To exploit environmental statistical regularities, the brain integrates information from various sensory modalities, enhancing the precision, accuracy, and processing speed of perception[1–6]. Temporal delays between information from the different sensory modalities provide a crucial cue for determining which cross-modal signals belong together[7,8]. However, the relative timing of signals from an external event can vary significantly between sensory modalities. The difference in velocities of light and sound introduces a lag for auditory signals relative to visual information. Since the speed of light and sound through air are constant, their time difference can provide absolute information about the source distance, independent of the environment. Indeed, in humans, audio-visual (AV) temporal disparity can influence distance estimation[9,10]. In addition to temporal disparities arising from physical variables in the external world, the temporal characteristics of stimuli that activate peripheral receptor cells differ considerably between sensory systems. As a result, the timescales of signals can differ by orders of magnitude across sensory systems. Thus, multisensory integration requires the comparison and linking of information across different modalities and timescales[11].

The superior colliculus (SC) receives sensory information from multiple modalities and is critical for reorienting an organism towards objects of interest and controlling innate behaviours[12–17]. Multisensory neurons in the SC respond to combinations of visual, auditory, and/or somatosensory stimuli, with their receptive fields (RFs) arranged to form topographically aligned spatial maps[18–20]. Interestingly, visual-auditory neurons in the SC show nonlinear integration of cross-modal information[21,22] and enhanced responses at variable delays between

[1]Neuronal Circuits and Behaviour Laboratory, The Francis Crick Institute, London, UK. [2]The Making lab, The Francis Crick Institute, London, UK. ✉e-mail: Florencia.iacaruso@crick.ac.uk

visual and auditory stimuli[23,24]. Such temporal integration may vary along the SC's topographic map, reflecting functional specialisations for distinct behavioural responses. Rodent studies have revealed that gene expression varies along the different anatomical axes of the SC[12] and show functional and anatomical segregation of inputs and outputs[12,25–28]. The medial SC encodes information about the upper visual field and promotes defensive responses, while the lateral region encodes information about the lower visual field and evokes approach and appetitive behaviour[25–28]. Whether these anatomically distinct regions of the SC exhibit specialisation in multisensory temporal integration to facilitate different behavioural responses remains unexplored.

To investigate how the SC represents AV temporal disparities, we recorded activity from approximately 5000 neurons across the SC of awake mice while presenting spatially coincident AV stimuli with varying temporal delays. We found that individual neurons exhibit a broad tuning to AV delays and nonlinear summation of unisensory inputs. At the population level, decoding accuracy for AV delays was highest in the postero-medial SC, which encodes the peripheral visual field. This enhanced temporal discriminability relies on the nonlinear integration of multisensory inputs in single neurons. Cross-correlation analysis revealed that spatial RF correlation is a strong determinant of connectivity in the SC and that multisensory neurons might receive a significant proportion of local input from other multisensory neurons. This suggests that recurrent local connections might contribute to nonlinear integration of cross-modal inputs across different time-scales. Our results suggest that region-specific functional specialisation in the SC underlies multisensory temporal integration, enhancing the encoding of AV information.

## Results

### Spatially aligned visual and auditory receptive fields in the mouse superior colliculus

To study the multisensory representation of space in the mouse SC, we recorded the activity of neuronal populations at a range of tissue depths and anatomical locations by performing extracellular electro-physiological recordings using Neuropixels 1.0 probes[29] in head-fixed awake mice (Fig. 1A). Recognising that sensory processing varies across different regions of the SC[30,31] we performed a comprehensive characterisation of AV integration by recording from different sites along the antero-posterior (AP), medio-lateral (ML) and dorso-ventral (DV) axes. We covered 1.7 mm, 1.9 mm, and 2.3 mm along the AP, ML, and DV axes of the SC, respectively. Anatomical locations of recording sites were confirmed by combining electrophysiological features with histological reconstructions of fluorescently-labelled probe tracks. We used a custom-made hemispheric stimulation apparatus to provide spatially coincident visual and auditory stimuli from a range of spatial locations (Fig. 1B), spanning 162° in azimuth and 72° in elevation across recordings (see 'Methods'). The stimuli consisted of a 100 ms white noise (1–22 kHz) and a 100 ms light flash presented via a dimmable light-emitting diodes (LED) positioned at the centre of each speaker. Sound level was fixed at 60 dB to enhance multisensory interactions at lower stimulus intensities[21] and to minimise potential confounds from sound-evoked body motion affecting auditory responses (Supplementary Fig. 1)[32,33]. We found that 16% of the recorded neurons responded to visual stimuli, 14% to auditory stimuli and 26% were classified as multisensory (Fig. 1C). Multisensory neurons exhibited a variety of response patterns, including neurons responsive to both modalities when presented independently (bimodal, 10%) and neurons responsive only to multisensory stimulation (gated, 16%). These neuronal subclasses were differentially distributed along the depth of the SC ($P = 1 \times 10^{-241}$, mixed-effects ANOVA; Fig. 1D). Consistent with previous reports[22,34], visual neurons were primarily located in the superficial layers (mean depth $756 \pm 318$ μm), while auditory neurons were found in the deeper layers of the SC ($1388 \pm 401$ μm). Bimodal neurons

were located more superficially ($992 \pm 458$ μm), whereas gated neurons were predominantly situated in the deeper layers of the SC ($1330 \pm 535$ μm).

To verify that our stimulation apparatus could capture the topographic alignment of visual and auditory maps in the SC[22,34], in a subset of experiments, we estimated the location of the centre of the RF of individual neurons by stimulating at 7 azimuths and 5 elevations with 18° steps surrounding the centre of the population RF, determined online. We observed that the RF centroid of visual and auditory responsive neurons recorded simultaneously with a vertical insertion of the Neuropixels probe were spatially aligned (Fig. 1E, F). As a result, the RFs obtained from the averaged population responses to visual and auditory stimuli of simultaneously recorded neurons were also aligned (Fig. 1G, H, $N = 11$ recordings, Pearson's correlation coefficient Azimuth: $R = 0.86$, $P = 1 \times 10^{-4}$; Elevation: $R = 0.76$, $P = 0.1$). Moreover, bimodal neurons exhibited overlapping unisensory visual and auditory RFs in azimuth (Fig. 1I, J; $n = 89$ neurons, Azimuth: $R = 0.4$, $P = 1 \times 10^{-4}$; Elevation: $R = 0$, $P = 0.9$). Additionally, for these neurons the RFs determined from multisensory stimulation were spatially aligned to the RF estimated from the average responses to the unisensory stimuli (Fig. 1K, Azimuth: $R = 0.56$, $P = 1 \times 10^{-9}$; Elevation: $R = 0.24$, $P = 0.02$), and the average population RF estimated from unisensory responses matched the multisensory RF (Fig. 1K, L, Azimuth: $R = 0.87$, $P = 1 \times 10^{-4}$; Elevation: $R = 0.69$, $P = 0.01$). Overall, these results confirm the spatial alignment of visual, auditory and AV RFs in the mouse SC, mapped through unisensory and multisensory stimulation. Moreover, RF location of visually and auditory responsive neurons was correlated with their anatomical location along the anterior-posterior axis of the SC (Supplementary Fig. 2).

### Temporal disparities between audio and visual inputs are encoded in superior colliculus neurons

To investigate how intermingled SC populations integrate AV information across different timescales, we presented auditory and visual stimuli at congruent spatial locations while systematically varying their temporal relationship. To determine the range of temporal disparities to test, we first studied the latency of the neuronal responses to unisensory visual and auditory stimuli. While we found maximum responses in visual neurons evoked by either the start (onset) or the end (offset) of the visual stimulus, auditory neurons primarily responded to the onset of the auditory stimulus (Fig. 2A). The difference in response onset between visual and auditory neurons was approximately 50 ms (Fig. 2A, median visual onset latency = 69 ms, interquartile range (IQR) = 35 ms; median auditory onset latency = 21 ms, IQR = 22 ms, $P = 1 \times 10^{-3}$, hierarchical bootstrap). This difference in response latency could result in a maximal multisensory integration when the visual stimulus precedes the auditory by this amount of time. We therefore presented unisensory visual and auditory stimuli, with varying AV delays ranging from 0 to 100 ms, ensuring that the visual stimulus always preceded the auditory one (Fig. 2B). Importantly, sound evoked uninstructed body movements were similar across all multisensory trial types ($P = 0.7$, Kruskal–Wallis, Supplementary Fig. 1).

Overall, we recorded a total of 5360 neurons from 24 mice (92 recording sites) and assessed their responsiveness using a ZETA test[35]. We found that 25% of the multisensory neurons (corresponding to 12% of the total recorded population) were modulated by AV delay (see 'Methods'). Within the multisensory population, AV delay-selective neurons constituted 23% of the bimodal group and 26% of the gated group. For all neurons that showed statistically significant differences between their multisensory and unisensory responses, we determined their preferred delay as the condition that evoked the highest peak firing rate (FR) (see 'Methods'). Overall, AV delay-selective neurons preferring different delays were intermingled with other neuronal groups, and there was no particular bias in their distribution along the different anatomical axes of the SC (Fig. 2C, mean depth $382 \pm 143$ μm,

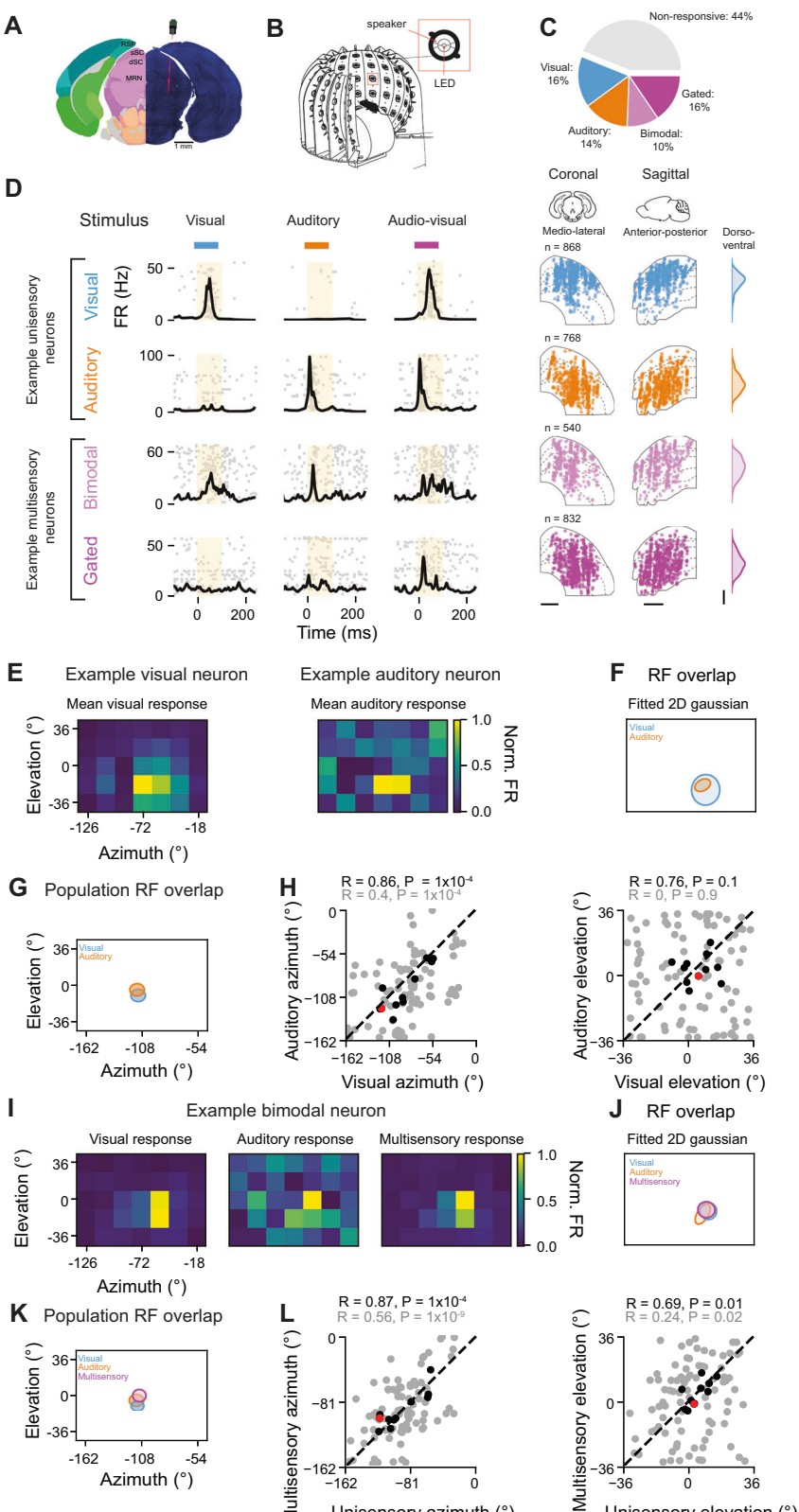

Spatial Autoregressive Model, SAM: $P = 0.1$; mean Moran Index $= 0$). Figure 2D, E shows the peristimulus time histograms and corresponding tuning curves for two example neurons. Consistent with previous reports[23,24], across the population of SC neurons the mean preferred AV delay was $54 \pm 30$ ms (Fig. 2F), which falls near the centre of the tested range (0–100 ms) and may partly reflect the distribution of delays tested. However, the distribution was broad, and we found neurons preferring each of the tested AV delays.

To characterise the selectivity of neuronal responses to specific AV delays, we calculated a specificity index (Fig. 2G, see 'Methods') and found that nearly half of the delay-selective neurons (48%) exhibited a high specificity index (>0.9). The degree of FR modulation by AV delay, was characterised using a FR modulation index (Fig. 2H, see methods). Most neurons showed a modest increase in FR at the preferred delay condition compared to the delay condition evoking the weakest response, with only 9% of neurons exhibiting an index equal to or

**Fig. 1 | Spatially aligned visual and auditory receptive fields in the mouse superior colliculus. A** Histological reconstruction of the probe track with DAPI (blue) and DiI (red) for an example recording site and corresponding alignment to the Allen Mouse Brain Atlas, mouse.brain-map.org[114]. RSP retrosplenial cortex, sSC superficial layers of the superior colliculus, dSC deep layers of the superior colliculus, MRM midbrain reticular nucleus. **B** Schematic of the experimental setup. Design obtained with Autodesk Inventor. **C** Proportion of neurons recorded in the SC, subdivided into non-responsive, visual, auditory, bimodal and gated. **D** Left: Response profiles for example unisensory and multisensory neurons. The spike raster plots for 50 repetitions are overlaid with the peristimulus time histogram (PSTH) during visual (blue), auditory (orange) and simultaneous audiovisual (AV) stimulation (magenta). Right: Anatomical location of all recorded neurons from each subclass along the medio-lateral (ML, left) and antero-posterior (AP, centre) axes of the SC. Distribution along the dorso-ventral (DV) axis is shown as a darnel density estimate plot (right). Scale bar: 0.5 mm. **E** Spatial heat map of normalised firing rates of two simultaneously recorded visual (left) and auditory (right) unisensory neurons. **F** Contours of 2D Gaussian fits for the RFs of neurons shown in (**E**). **G** Averaged population visual and auditory RFs from an example recording. Size of the ellipses denote 90% confidence intervals around the mean RF coordinates. **H** Correlation between visual and auditory RF azimuth (left) and elevation (right) for individual neurons (grey circles) and individual recordings (black circles). Red circle corresponds to the example recording shown in **G**. $N = 9$ recordings, Pearson correlation coefficient (R) and two-sided $p$ value from Pearson's correlation test are provided. **I** Same as **E** for an example bimodal neuron for visual (left), auditory (centre) and multisensory (right) stimulation. **J** Overlap between visual, auditory and multisensory RFs of neuron in (**I**). **K** Same as **G** for population of bimodal neurons. **L** Correlation between multisensory and unisensory azimuth (preferred unisensory modality determined independently for each neuron, left) and elevation (right) for individual neurons (grey circles) and for the population (black circles). Red dot corresponds to the example recording shown in (**K**). Pearson correlation coefficient (R) and two-sided $p$ value from Pearson's correlation test are provided.

exceeding 0.5 (median = 0.28, IQR = 0.17). Furthermore, the reliability of responses at the preferred delay was relatively low, with only 7% of neurons showing a high trial-to-trial correlation coefficient (>0.3, median = 0.10, IQR = 0.09; Fig. 2I). Nonetheless, the reliability of individual neurons across trials was higher than chance for 99% of the delay-selective neurons (chance median = −0.0002, IQR = 0.003, see 'Methods'). This demonstrates that the multisensory responses were reliable and specific to AV delays. Reliability was comparable across the different AV delays tested (FR modulation index: $P = 0.54$; reliability index (RI): $P = 0.68$, Linear mixed effects model, LMM), while selectivity index was weakly different between delays (selectivity index: $P = 0.003$, LMM; slope: −0.067 selectivity index units/100 ms audio-visual delay). Together, these results show that a wide range of AV delays are represented in the responses of individual SC neurons, with a high degree of specificity.

Given the similarity of the mean preferred AV delay across the population and the difference in the mean auditory and visual latencies, we sought to determine whether AV delay selectivity could be predicted from responses to unisensory inputs at single neuron level. However, as previously reported[24], the preferred AV delay of individual neurons was not simply predicted by the difference in latency between their responses to unisensory stimuli (Fig. 2J, $P = 0.97$; LMM), nor by their latency of the unisensory responses alone (Supplementary Fig. 3, visual latency: $P = 0.77$; auditory latency: $P = 0.83$, LMM). To assess if the temporal dynamics of the unisensory response profiles are indicative of AV delay selectivity, we linearly added the FRs of the visual and the auditory response, after shifting the latter by the different delays experimentally tested, and estimated the peak FR of these summed AV responses. The AV delay with maximum peak FR from the linearly summed responses was selected as the predicted preferred delay of the neuron (see 'Methods', Fig. 2K). We found no significant relationship between the observed and predicted preferred delay ($P = 0.49$, LMM) and overall, the preferred delay was higher than the predicted AV delay (mean ± s.d., observed: $54 \pm 30$ ms; predicted: $41 \pm 32$ ms; $P = 0.03$, hierarchical bootstrap, Fig. 2L). Interestingly, discrepancies between the predicted and observed responses revealed nonlinearities in the summation of auditory and visual inputs. To characterise these nonlinearities across AV delays, we calculated the multisensory interaction index (MII, see 'Methods'). Nonlinear interactions were classified as supralinear when the multisensory response exceeded the sum of unisensory responses and sublinear when it fell short. Specifically, neurons that exhibited a 50% increase in FR relative to the sum of unisensory responses (MII > 0.5) were classified as exhibiting supralinear interactions, while those with a 50% reduction (MII < −0.5) were classified as sublinear.

Linear summation characterised the response of most neurons when auditory and visual stimuli were presented simultaneously ($AV_0$, median MII = 0.00, IQR = 0.33, Fig. 2M), while a small proportion of neurons displayed supralinear (8.9%) or sublinear (1.2%) multisensory responses (Fig. 2N). Mean MII across the population of delay tuned neurons was similar across the delays tested ($P = 0.8$, LMM). When considering only the preferred delay for each neuron, 33% of neurons exhibited supralinear summation of unisensory inputs (median MII at preferred delay = 0.33, IQR = 0.59, Fig. 2N). Importantly, in a subset of experiments we studied if the order of presentation of auditory and visual stimuli affected nonlinear interactions. We found that supralinear summation was weaker when auditory stimuli preceded visual stimuli (median MII auditory leading = 0.02, IQR = 0.50, median MII visual leading = 0.15, IQR = 0.41; $P = 0.03$, LMM; Supplementary Fig. 4). This suggest that the temporal window for multisensory integration is asymmetric and might be best matched to represent visual information followed by auditory. Overall nonlinear multisensory interactions were observed throughout the different anatomical axes of the SC. The mean absolute MII, which accounts for supra- and sublinear interactions, was larger in the deeper layers of the SC (ML: $P = 0.9$; AP: $P = 0.2$; DV: $P = 1 \times 10^{-8}$; LMM, Supplementary Fig. 5). Importantly we did not observe any clustering of supralinear or sublinear responses (Supplementary Fig. 5). These results indicate that, as in other species[36,37], mouse SC neurons exhibit nonlinear summation of AV information[22,34].

## Functional organisation of multisensory networks in the superior colliculus

To study if the functional network organisation of the SC could contribute to the nature of multisensory responses, we performed cross-correlation analysis between simultaneously recorded neuronal pairs along the DV axis[38–40]. This approach allowed us to study putative connections across the layers of the SC. To exclude stimulus evoked responses, we calculated the cross-correlogram (CCG)[41,42] on spike trains recorded during the inter-trial interval. Only responsive neurons were included in the analysis to avoid spurious correlations among unresponsive neurons ($N = 3543$ responsive neurons, 92 recordings, 24 mice). The approach used[39] identifies positive peaks in jittered-corrected CCGs, to correct for slow timescale correlations and excludes significant inhibitory correlations (see 'Methods', Supplementary Fig. 6). Putative connections were defined by lags of 1–5 ms, connection directionality was obtained from the sign of the lag, and the strength of the interaction was estimated from the peak of the CCG. Overall connection percentage was 2.3% (range: 0.2–10.7%, 2.3% vs false positive rate = 0.9%, $P = 1 \times 10^{-10}$, Mann–Whitney U test) and decreased exponentially with increasing neuronal pair distance in the DV axis (Fig. 3A). Interestingly DV connection percentage decayed along the ML axis of the SC with a higher percentage of connection for recordings from the medial SC compared to the lateral SC (ML: $P = 1 \times 10^{-5}$; AP: $P = 0.2$, LMM; Fig. 3B). This spatial gradient was not observed when analysing pairs of correlated neurons with 0 ms lag, that could reflect shared inputs instead of putative connections (ML:

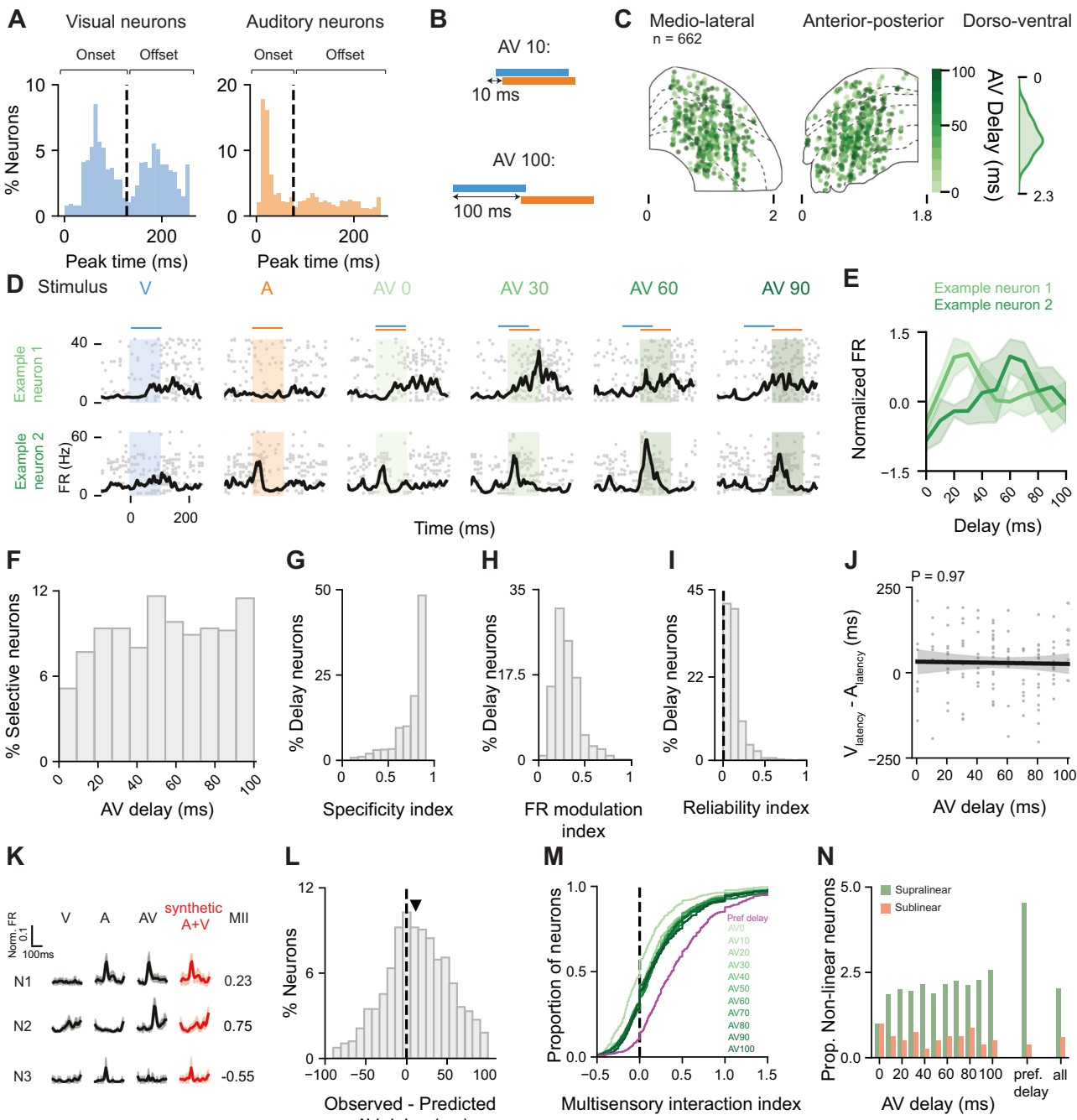

**Fig. 2 | Temporal delay of audiovisual information is encoded in superior colliculus neurons. A** Distribution of response latency for visual (blue left) and auditory (orange right) unisensory responses. **B** Schematic of stimulation protocol. **C** Anatomical distribution of neurons modulated by AV delay. Left: ML; Centre: AP; Right: DV axes. **D** Raster plots and PSTHs for two example neurons to unisensory and multisensory stimuli presented at variable delays. **E** AV delay tuning curve for neurons shown in **D**. Data are represented as mean values ± SEM. **F** Distribution of preferred AV delay across the population of delay-selective neurons. Distribution of specificity index (**G**); firing rate modulation (**H**) and reliability index (**I**) for delay-selective neurons. **J** Correlation between the neurons' preferred AV delay and the difference in latency between visual and auditory response peak time. Only delay neurons that were responsive during both visual and auditory stimulation were included ($N = 123$). $P$ value from two-sided LMM. The bold line represents the fitted regression (mean predicted values), while the shaded grey area denotes the 95%

confidence interval. **K** Unisensory and multisensory responses for three example neurons and corresponding synthetic multisensory response obtained from the addition of auditory and visual instantaneous firing rates. Responses are represented as mean values ± SEM. Numbers indicate multisensory interaction index (MII). **L** Distribution of difference between observed and predicted AV delay preference for AV delay-selective neurons. **M** Cumulative distribution of multisensory interaction index. Green tone indicates AV delay; magenta corresponds to MII calculated at the preferred delay of each individual neuron. **N** Proportion of neurons exhibiting supra- and sublinear multisensory interaction, normalised by the number of neurons exhibiting supra- or sublinear multisensory interactions at AV delay 0 ms (supralinear = 8.9%; sublinear = 1.2%). Proportions are shown for each AV delay tested, as well for the preferred delay condition and as the mean value across all conditions (all).

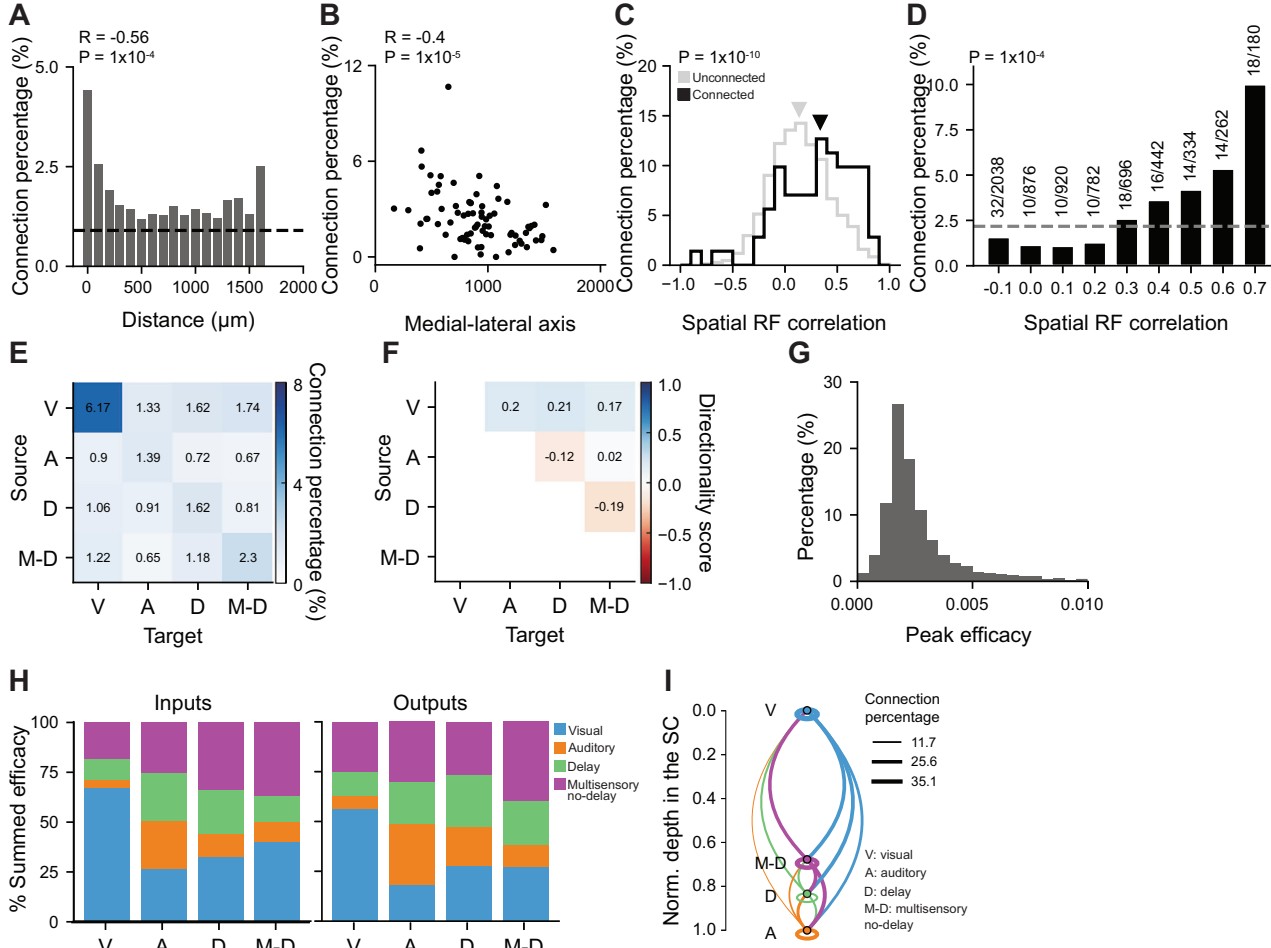

**Fig. 3 | Functional specificity of dorso-ventral connectivity within the SC.**
**A** Mean putative connection percentage as a function of neuronal pair distance along the DV axis (step size 100 μm; $N = 1421$ putative connections over 69044 assessed, 92 recordings, 24 mice). Black dashed line represents the false positive rate estimated from applying cross correlation analysis to independently recorded neurons. Pearson correlation coefficient (R) and two-sided $p$ value from Pearson's correlation test are provided. **B** Connection percentage as a function of mean location along the medio-lateral axis. Each dot represents one recording. Pearson correlation coefficient (R) and two-sided $p$ value from Pearson's correlation test are provided. **C** Distribution of pairwise spatial multisensory receptive field correlation for connected (black) and unconnected (grey) neuron pairs. Arrowheads indicate the medians of the distributions. Only sensory-responsive neurons were included ($N = 246$ neurons, 11 recordings, 9 mice). $P$ value from a two-sided

Kolmogorov–Smirnov test. **D** Relationship between connection percentage and pairwise multisensory spatial receptive field correlation. Grey dashed line indicates mean connection percentage. $P$ value from a two-sided permutation-based Cochran–Armitage trend test. **E** Connectivity matrix between different subpopulations of neurons: V: visual, A: auditory, D: delay-selective, M-D: multisensory excluding delay selective neurons. **F** Directionality matrix as in **E**. **G** Distribution of excitatory putative connection strength. **H** Percentage of the total summed efficacy for inputs (left) and outputs (right) for different functional subpopulations. **I** Schematic representation of functional connectivity observed in the DV axes of the SC, based on (**H**). Neuronal subpopulations are represented by circles, and are distributed according to their mean location along the DV axis of the SC. The line thickness indicated the total summed connection efficacy. Connections within functional subpopulations are shown as ellipses.

$P = 0.25$; AP: $P = 0.42$, LMM; Supplementary Fig. 6). This suggests that shared input and recurrent connectivity contribute differently to multisensory integration across SC regions.

Moreover, we investigated whether spatial RF similarity predicts the likelihood of functional connectivity, as previously shown in visual cortex (Cossell et al.[43]; Iacaruso et al.[44]). To test this, we performed CCG analysis on a subset of experiments in which fine spatial RFs were mapped (corresponding to Fig. 1). For this analysis, only sensory-responsive neurons were included ($N = 246$ neurons, 11 recordings, 9 mice). In order to address the relevance of spatial coincidence for functional connectivity, we focused on the analysis of multisensory spatial RFs, when both visual and auditory stimuli were presented simultaneously. This analysis revealed that putatively connected neuron pairs exhibited significantly higher spatial signal correlation compared to unconnected pairs ($P = 1 \times 10^{-10}$, Kolmogorov–Smirnov test; Fig. 3C). Furthermore, connection probability increased with increasing spatial RF correlation and was significantly different to the

session-shuffled null distribution ($P = 1 \times 10^{-4}$, permutation-based Cochran–Armitage trend test; Fig. 3D), indicating that neurons with more similar spatial tuning are more likely to be functionally connected. These results demonstrate that spatial RF similarity is a strong predictor of functional connectivity in the SC, extending a key organisational principle previously described in the cortex to subcortical sensory circuits.

We then assessed connection probability and directionality between previously defined neuronal subpopulations (Fig. 3E, F), returning to our larger dataset on AV temporal modulation ($N = 3543$ responsive neurons, 92 recordings, 24 mice). This analysis revealed a high degree of functional specificity in the network connectivity. The percentage of connections between visual neurons was 6.17%, while auditory neurons received the highest percentage of connections from other auditory neurons, albeit their connectivity rate was much lower (1.39%). Importantly, the remarkably higher connectivity of visual neurons compared to auditory neurons was not explained by a

difference in FR during the inter-trial period used to estimate connectivity (Supplementary Fig. 6). Overall, visual neurons provided most of the inputs to other neurons in the network, with the only exception of auditory neurons, consistent with the prominent visual role of the SC (Fig. 3E). Interestingly, multisensory neurons preferentially connect within their own subtype and when grouping together multisensory neurons (AV delay-selective and delay-nonselective neurons) they were more strongly connected with other multisensory neurons than to unisensory neurons (Fig. 3E, F). However, connectivity between delay-selective neurons was not explained by their preferred delay similarity (Supplementary Fig. 6). Overall, the pattern of connectivity between functional subclasses was consistent across the ML axes of the SC albeit with a lower rate of connectivity in the lateral SC for all the functional subpopulations and in particular for delay-selective neurons (Supplementary Fig. 6).

To further characterise network interactions, we used the interaction strength as a proxy for putative connection strength. The distribution of interaction strength was highly skewed (median interaction strength: 0.002; s.d.: 0.05; Fig. 3G) with few strong interactions among a majority of weaker ones. To simultaneously account for interaction strength and interaction probability, we quantified the total sum of the interaction strength received by each neuronal functional subpopulation by combining data from all interacting pairs (Fig. 3H). This analysis allowed us to build a model of putative DV connectivity of the SC (Fig. 3I) and revealed that overall multisensory neurons (AV delay-selective and delay-nonselective neurons) receive less than 50% of the total local interaction strength from SC unisensory neurons, with the rest of the input arising from other multisensory neurons potentially contributing to their nonlinear multisensory integration.

## Distinct contributions of SC subpopulations to the temporal representation of multisensory information

The inter-trial variability (Fig. 2J), and the relatively low response modulation evoked by specific AV delays in single SC neurons (Fig. 2I), together with the widespread connectivity between multisensory neurons (Fig. 3F, G) suggest that precise decoding of a wide range of AV delays rely on a distributed representation, i.e., from the collective activity pattern of the population. To assess how accurately AV delay can be predicted from population responses, we performed cross-validated, multi-class classification of single-trial responses using a linear classifier support vector machine, (SVM) and evaluated its performance on held-out data (80% training set, 20% test set). The classifier was trained to decode the 11 tested multisensory stimuli, as well as unisensory auditory and visual trials. We controlled for the lagged presentation of the auditory stimuli in multisensory trials by adding 10 synthetic unisensory auditory trials shifted in time (Fig. 4A). This approach allowed us to distinguish the classifier's accuracy in decoding AV delays from its ability to determine the timing of a unisensory auditory response. The classifier achieved an 85% accuracy rate in decoding delay conditions when all neurons were included, outperforming chance levels ($N = 5360$ neurons; $P = 1 \times 10^{-8}$, Mann–Whitney U test, Fig. 4B, C). Our analysis shows that decoding the range of AV delays used in this study can be achieved with as few as 9 neurons sampled randomly from all SC subregions, resulting in a decoding accuracy significantly above chance level (>9.09% decoding accuracy, $P = 1 \times 10^{-6}$, Wilcoxon signed-rank test, Fig. 4D). Importantly we could not decode the AV delay from analysis of body movement (Supplementary Fig. 7).

This result demonstrates that the SC can encode the precise temporal disparity between the onset of visual and auditory stimuli with a relatively low number of neurons. Furthermore, we observed a lower performance in classifiers trained on simultaneously recorded neurons compared to those trained on random samples of neuronal responses obtained across 92 recordings ($N = 25$ neurons per model;

median accuracy for randomly sampled neurons = 19%, IQR = 10%; median accuracy for simultaneously recorded neurons = 11, IQR = 14; $P = 0.003$, Mann–Whitney U test, Supplementary Fig. 8). This result suggests that noise correlations, amplified by recurrent connectivity, may constrain distributed population coding[45,46]. It might also indicate that variability on the representation of sensory information within the recorded population might contribute to the observed differences in performance[47].

To characterise the contribution of different neuronal populations to the decoding of AV delay, we compared the decoding accuracy of visual, auditory, multisensory delay-selective and multisensory delay-nonselective neurons, randomly sampling groups of 200 neurons from these populations. As before, auditory shifted trials were included as a control for each classifier. Fig. 4E shows the confusion matrix for each group. This analysis revealed differences in AV delay decoding accuracy between subpopulations of neurons ($P = 1 \times 10^{-15}$, Kruskal–Wallis, Fig. 4F). Auditory, multisensory delay-selective and multisensory delay-nonselective neurons showed the highest decoding accuracy (Fig. 4F, auditory vs. visual neurons $P = 1 \times 10^{-8}$; delay-selective vs. visual neurons $P = 1 \times 10^{-8}$; delay-nonselective vs. visual neurons $P = 1 \times 10^{-8}$, Mann–Whitney U test, Benjamini–Hochberg multiple comparison). Importantly, despite the high accuracy obtained for auditory neurons, this group also displayed a high rate of misclassification between AV delay trials and synthetic shifted auditory trials (Fig. 4G). In contrast, the low decoding accuracy of visual neurons, explained by the prediction of multisensory responses at an incorrect AV delay, suggests that their responses are weakly modulated by the delayed presentation of an auditory stimulus. As expected, the highest accuracy was observed by multisensory neurons previously labelled as AV delay-selective (Fig. 4F). The accuracy of the classification quantifies the rate of correct classifications. We next sought to quantify the precision of the classification for the different subpopulations. We therefore calculated the difference between the AV delay estimated from each trial classification and the actual AV delay for which the responses were measured. We quantified the classification precision of multisensory AV delay-selective and delay-nonselective neurons and found that for classifiers trained on the activity of 200 AV delay-selective neurons and tasked to decode AV delays ranging from 0 to 100 ms, separated by 10 ms intervals, the precision was ~2 ms (median ± IQR: AV delay-selective = 2 ± 1 ms; multisensory AV delay-nonselective = 6 ± 2 ms, $P = 1 \times 10^{-8}$, Kruskal–Wallis, Fig. 4H).

To verify the contribution of different subpopulation to the decoding of AV delays within individual recordings, we analysed the decoding accuracy obtained from models trained on the responses of 25 randomly selected simultaneously recorded neurons using a LMM ($N = 70$ recordings, 20 repetitions). High decoding accuracy was mostly attributed to the proportion of delay neurons present in the individual recordings and was not explained by the proportion of multisensory delay-nonselective neurons in the recording (multisensory delay-selective: $P = 1 \times 10^{-9}$, visual: $P = 0.03$, auditory: $P = 1 \times 10^{-6}$, multisensory delay-nonselective neurons: $P = 0.4$, linear mixed model (LMM), Fig. 4I, L). Interestingly, despite the poor decoding performance of the visual subpopulation, the proportion of visual neurons in individual recordings was significantly correlated with decoding performance. This suggests a potential contribution of visual neurons when intermingled with other subpopulations by stabilising baseline population activity. Together these results highlight the contribution of AV delay-selective neurons to the accurate temporal representation of multisensory information.

## Nonlinear summation of auditory and visual inputs improves the temporal precision of multisensory representations

AV delay selective neurons exhibit strong nonlinear integration of audio-visual signals (Fig. 2N). We next sough to determine the extent to which nonlinear summation of sensory information contributes to the

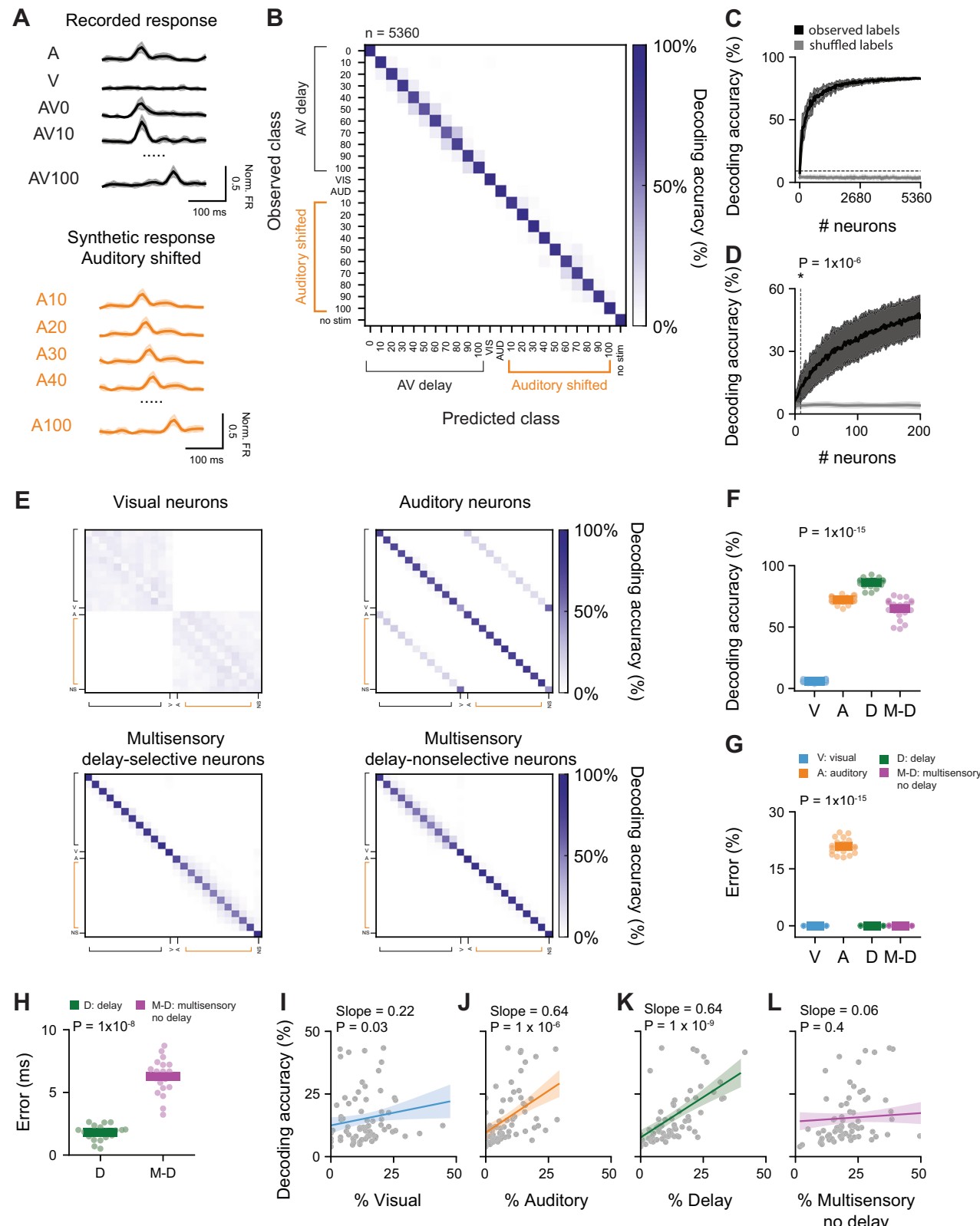

discrimination of AV delays at the population level. We compared two independent linear classifiers: one trained on the measured multisensory responses and the other on synthetic AV responses obtained by linearly adding unisensory visual and auditory responses, with the latter shifted by each of the temporal delays tested experimentally. As before, unisensory visual, auditory and auditory-shifted trials, along with a blank trial, were included as a control in both classifiers. When

considering multisensory delay-selective neurons, the classifier trained on the measured multisensory responses outperformed the classifier trained on the synthetic linear responses (performance for 200-neuron classifier on observed nonlinear responses = 86 ± 4% vs synthetic linear responses = 6 ± 2%, $P = 1 \times 10^{-8}$, Mann–Whitney U test, Fig. 5A). Similar results were obtained when including a sample of neurons from the whole population, suggesting that small

**Fig. 4 | Population encoding of audiovisual delay in the superior colliculus.**
**A** Example trials used to train a support vector machine (SVM) classifier. Neuronal responses aligned to the auditory stimulation during multisensory trials (orange traces) were included as a control for the classification of the timing of auditory stimulation. Responses are represented as mean values ± SEM. **B** Confusion matrix obtained from linear classification of AV delay trials using 5360 neurons. Auditory trials shifted in time are included in the classification (orange labels). **C** Mean decoding accuracy of linear classifier for AV delay trials as a function of number of neurons included in the classifier (black). The classifier trained on shuffled labels is shown in grey. The shaded area represents the standard deviation over 20 repetitions. The dotted line represents chance level. **D** Decoding accuracy plotted as in (**C**), for up to 200 neurons across 200 iterations. The dotted line represents the number of neurons necessary to achieve higher than chance performance. $N = 9$, $P$ value from a two-sided Wilcoxon signed-rank test. **E** Same as **B** for 200 neurons randomly sampled from visual (V), auditory (A), delay-selective (D) or multisensory non delay-selective (M-D) neurons. **F** Mean decoding accuracy calculated from the

confusion matrices in **E**. P value from a two-sided Kruskal–Wallis test. **G** Misclassification error rate (%) between AV delay trials and synthetic shifted auditory trials calculated for 200 randomly sampled neurons from different neuronal subpopulations. $P$ value from a two-sided Kruskal–Wallis test. **H** Misclassification error (ms) for 200 randomly sampled D and M-D neurons for classifiers only trained in AV delay trials. $P$ value from a two-sided Kruskal–Wallis test. **I–L** Linear decoders were obtained from individual recordings with at least 25 neurons. Partial dependence plots show mean classifier decoding accuracy as a function of the percentage of visual (**I**), auditory (**J**), delay-selective (**K**) and multisensory delay-nonselective (**L**) neurons. The bold line represents the fitted regression (mean predicted values). The gray area denotes the 95% confidence interval. Dots indicate values from individual recordings, $P$ values from two-sided linear mixed model. An 80% training and 20% testing split with 5-fold cross-validation was applied across all SVM implementations. All confusion matrices shown and the analysis of decoding accuracy and error correspond to the test set.

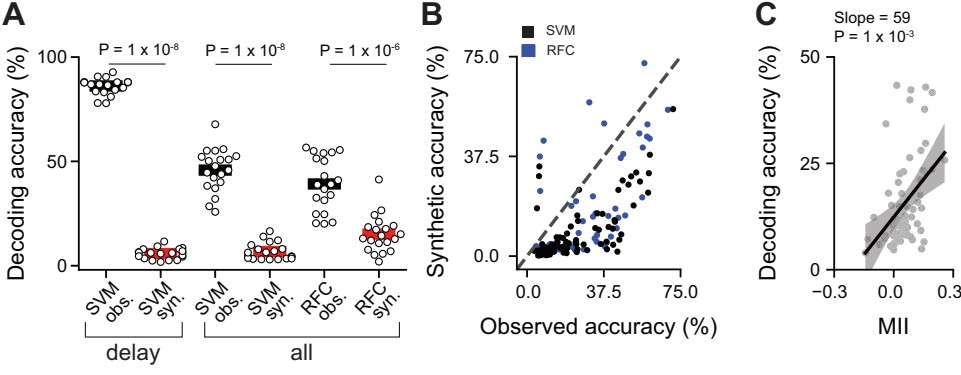

**Fig. 5 | Encoding of audiovisual delays is enhanced by nonlinear summation of unisensory inputs.** **A** Comparison of decoding accuracy between classifier trained on recorded responses (black) and classifier trained on synthetic responses (red). The classifiers were trained on the responses from 200 delay selective neurons (left) or randomly sample neuronal populations (right). $P$ values from a two-sided Wilcoxon signed-rank test. **B** Decoding accuracy for classifier trained on synthetic responses versus accuracy from classifier trained on observed responses, for individual recordings. Number of neurons included in the classifier varied for each recording. Range 11–130 neurons per recordings; $N = 92$ recordings. Black and blue

circles correspond to SVM and RFC, respectively. **C** AV delay decoding accuracy as a function of the mean MII of all neurons included in the analysis. Gray circles correspond to individual recordings. Recordings were included if they contained at least 25 neurons; decoding accuracy was calculated using a subset of 25 neurons per recording. The bold line represents the fitted regression line (mean predicted values), while the shaded gray area denotes the 95% confidence interval. $P$ value from two-sided linear mixed model. An 80% training and 20% testing split with 5-fold cross-validation was applied across all decoding implementations. The analysis of decoding accuracy corresponds to the test set.

nonlinearities found across the population can also contribute to the decoding of multisensory information (performance for 200-neuron classifier on observed nonlinear responses = 46 ± 10% vs synthetic linear responses = 7 ± 4%, $P = 1 \times 10^{-8}$, Mann–Whitney U test, Fig. 5A). Nonlinearities similarly increased accuracy when training a Random Forest Classifier (RFC), even when this classifier should be able to capture other nonlinear interactions in the population response (RFC; performance for 200-neuron classifier on observed nonlinear responses: 39 ± 13% vs synthetic linear responses 15 ± 9%, $P = 1 \times 10^{-6}$, Mann–Whitney U test, Fig. 5A). Similar results were obtained when comparing the performance of classifiers trained on individual recordings (Fig. 5B). In this case, the whole population was included given the small sample size found on individual recordings. Moreover, decoding accuracy on individual recordings was strongly correlated with the degree of nonlinearities in the recorded population ($P = 1 \times 10^{-3}$, Fig. 5C). These results demonstrate that nonlinearities in SC neuronal responses enhance the accuracy of the representation of AV information.

**Enhanced audiovisual delay encoding in the medial-posterior SC**
Previous studies have shown asymmetries in behavioural multisensory interactions attributed to the SC, with stronger AV integration in the temporal hemifield compared to the nasal hemifield[48]. Functionally

distinct regions of the SC exhibit unique patterns of inputs and outputs[27], as well as differential connectivity along the DV axis (Fig. 3B) that could give rise to differences in multisensory processing across the sensory field. These distinct local properties could explain the variability in AV delay encoding accuracy observed across individual recordings (Fig. 5C). Therefore, to investigate how population-level encoding of AV delays varies along the ML, AP, and DV axes of the SC, we trained the SVM classifier on individual recordings and assessed how AV delay decoding accuracy changed with the different axes within the SC. We found that decoding accuracy exhibits a gradient along the ML and the AP axes ($N = 70$ recordings, 25 neurons per recording, $P = 1 \times 10^{-3}$ multivariate linear regression, Fig. 6A, B). Similar results were obtained when training the SVM on randomly sampled neurons within spatial bins of 650 μm along each axis. Decoding accuracy varied along the three anatomical axes ($P$ value: ML = 0.02, AP = $1 \times 10^{-5}$, DV = $1 \times 10^{-4}$; Kruskal–Wallis test, Supplementary Fig. 9). Specifically, accuracy in the most medial portion of the SC was, on average, 11% higher than in the most lateral region ($P = 0.03$, Mann–Whitney U test, Benjamini–Hochberg multiple comparison, Supplementary Fig. 9). Similarly, accuracy in the posterior SC was 12% higher than in the anterior SC, and along the DV axis the deepest SC region showed 12% higher accuracy ($P$ value AP = $1 \times 10^{-4}$, DV = $1 \times 10^{-3}$, Mann–Whitney U test, Benjamini–Hochberg multiple comparison,

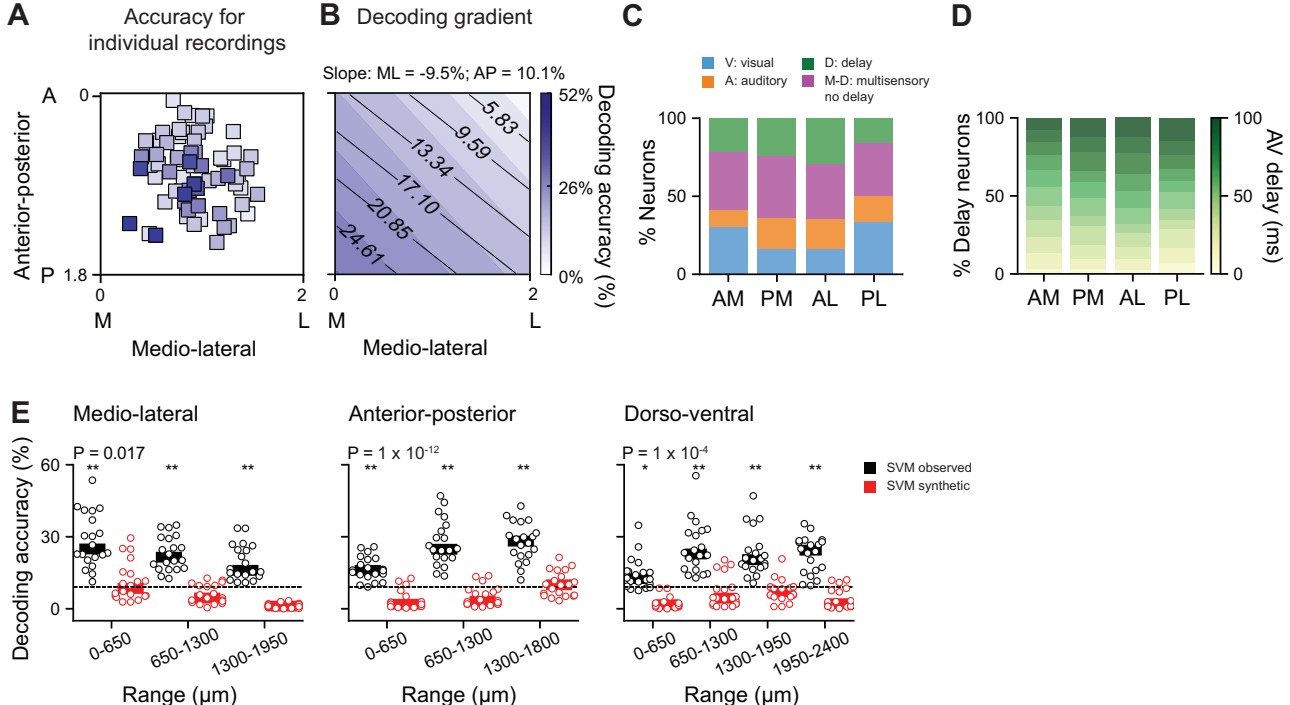

**Fig. 6 | Population decoding of audiovisual delays is enhanced in the posteromedial SC. A** Anatomical distribution of individual recordings colour coded according to the mean AV delay classifier accuracy obtained with 25 neurons, 20 repetitions. **B** Decoding gradient as a function of ML and AP location, calculated from the results shown in (**A**). **C** Percentage of visual (V), auditory (A), delay-selective (D) and multisensory non delay-selective (M-D) neurons across four anatomical bins: AM anterior-medial, PM posterior-medial, AL anterior-lateral, PL posterior-lateral. Each bin covered 2.20 mm³, with on average 1039 ± 121 neurons per bin. **D** Percentage of neurons selective for different AV delays (0–100 ms) for the same anatomical bins as in (**C**). **E** AV delay decoding accuracy for 20 iterations

of the linear classifier trained on the responses of 50 randomly sampled neurons within 650 μm bins along the ML (left), AP (centre) and DV (right) axes of the SC. The classifiers were either trained on the observed responses (black) or on temporally shifted linearly summed audio visual responses (red). The dotted line represents chance level. **$P = 10^{-6}$; *$P = 10^{-4}$ with respect to chance, two-sided Wilcoxon signed-rank test. $P$ values indicate differences across anatomical locations for observed responses, two-sided Mann–Whitney U test. An 80% training and 20% testing split with 5-fold cross-validation was applied across all SVM implementations. The analysis of decoding accuracy corresponds to the test set.

Supplementary Fig. 9). These findings were robust across a range of bin sizes and neuron sample sizes (Supplementary Fig. 9). When considering all three anatomical axes simultaneously by randomly sampling 50 neurons from eight distinct areas of the SC, each covering a volume of 1.09 mm³ (528 ± 243 neurons per bin), the accuracy remained higher in the posteromedial region ($P = 1 \times 10^{-16}$, Kruskal–Wallis, Supplementary Fig. 9).

To understand if the difference in decoding accuracy could be explained by a differential representation of neuronal subpopulations, we compared the proportion of visual, auditory, multisensory delay-selective and delay-nonselective neurons across the different anatomical axes of the SC. The SC was divided into four subregions, each covering 2.20 mm³ (1039 ± 121 neurons per bin), and the distribution of neuronal subpopulations was compared across these regions. Our correlation analysis between decoding accuracy and the distribution of subpopulations did not reach significance for any of the subpopulations, albeit suggested a trend for auditory, delay-selective and delay-non selective multisensory neurons (for percentage of visual neurons: $P = 0.4$, auditory: $P = 0.06$, delay selective: $P = 0.1$, multisensory delay-nonselective neurons: $P = 0.1$, LMM, Fig. 6C). Additionally, the preferred delays of AV delay-selective neurons showed no differences across spatial bins ($P > 0.4$ paired comparisons across all subregion combinations, Kolmogorov–Smirnov test, Fig. 6D). Moreover, the observed regional encoding differences were maintained when training classifiers on subpopulations of visual, auditory, multisensory and delay-selective neurons, although decoding accuracy was similar for the latter group across the ML axis (Supplementary Fig. 10).

We next examined if the degree of nonlinear integration of multisensory information across anatomical regions might explain the gradient in decoding performance. The difference in AV delay population decoding across SC regions was not explained by differences in the averaged multisensory integration index across the population in specific SC subregions ($P = 0.7$, LMM). However, the mean multisensory index does not capture the diversity of multisensory interactions across the population. We therefore studied if nonlinear integration might contribute to the observed gradient in decoding of AV delays by training classifiers on synthetic nonlinear responses. The decoding accuracy decreased across all SC regions when training the SVM classifier with synthetic linear responses (Fig. 6E) and although a small gradient in decoding performance was still present, the decoding accuracy was not higher than chance level for any of the bins tested. Taken together these results show that SC regions encoding information from the upper-temporal visual field are best suited for representing temporal asynchronies in the onset of visual and auditory signals. This functional specialisation of multisensory temporal integration in SC posteromedial subregions likely arises from nonlinear integration of AV signals in mixed functional subpopulations specifically interconnected.

## Discussion

We investigated the temporal integration of auditory and visual information in the mouse SC at the single-neuron and population levels. A wide scope of previous work has shown that multisensory integration across brain regions depends on the temporal relationship of the sensory inputs[49–51]. We found neurons exhibiting tuning to AV

asynchronies that are widely distributed across the SC. The AV delay preference of individual neurons emerges from nonlinear integration of unisensory inputs, resulting in delay preferences longer than those predicted by linear summation of unisensory responses. These nonlinearities are more prevalent when visual information precedes auditory input, suggesting a mechanism that supports the efficient encoding of ethologically relevant multisensory information. Using population-level decoding via SVM classifiers, we show that nonlinear integration enhances the precision and accuracy of the SC's representation of AV delays. Furthermore, the medial-posterior SC, which represents the peripheral visual field, exhibited greater AV delay encoding accuracy compared to lateral and anterior regions of the SC. Functional connectivity analysis using cross-correlation revealed that SC neurons tend to form connections within their functional subclasses—visual, auditory, multisensory, or AV delay-selective neurons. This functional specificity suggests that SC connectivity is organised to optimise sensory integration while preserving the distinct processing roles of these subclasses.

## Nonlinear audiovisual integration in the mouse SC

Nonlinear AV integration in the mouse SC has been previously reported, with early auditory processing modulating later visual responses as a proposed mechanism for multisensory integration[22]. However, recent reports failed to find spatial auditory RFs and nonlinear summation of AV information in the mouse SC[52]. The authors suggested strong modulation of auditory neurons by body movements[52], highlighting the challenges of measuring auditory responses in behaving animals. To minimise the effects of uninstructed stimulus-evoked movements in our study, we recorded from habituated and passively stimulated mice using low sound intensity. The auditory origin of our recorded responses is supported by the short latency of auditory neurons, the specificity of AV delay neurons, and the absence of AV delay-specific triggering of uninstructed movements. Our findings are generally consistent with those previously reported by Ito et al.[22] who showed that the mouse SC contains topographically organised visual and auditory neurons that exhibit nonlinear multisensory integration. Their model-based analysis, based on the neuronal responses to a fixed AV delay, showed that multisensory integration is driven by an early auditory response modulated by a subsequent visual response. Our approach enabled the characterisation of AV integration in the temporal domain using a range of temporally shifted auditory and visual stimuli, which evoked stronger nonlinear integration when visual stimuli preceded auditory. We identified AV delay-selective neurons, a subset of bimodal and gated multisensory neurons, that exhibit nonlinear summation of sensory inputs at specific temporal disparities. Our results show that unimodal response latency is a poor predictor of audio visual delay selectivity, highlighting the relevance of exploring the multidimensional parameter space.

Selectivity of AV delay tuned neurons could contribute to the sparsity of multisensory representations in the SC, by selectively increasing FRs to specific combinations of AV stimuli while otherwise maintaining a relatively low FR. Nonlinear selectivity can increase the dimensionality of sensory representations, enhance stimulus discriminability, minimise errors, and provide a robust neural code[53-55],. Consistently, our results show that nonlinear multisensory integration improves the accuracy of AV delay decoding compared to linear models. This mechanism may facilitate the extraction of population responses by downstream regions[54], such as the lateral geniculate nucleus, lateral posterior nucleus of the thalamus, zona incerta and the parabigeminal nucleus[28,56], to support SC-mediated sensory and motor responses[57].

The cellular and circuit mechanism underlying multisensory responses in SC neurons are not yet fully understood. Modelling studies suggest that clustering of synaptic inputs may play a key role[58-60]. Indeed, multisensory responses in collicular neurons depend on NMDA-gated channels[61], suggesting that the spatial arrangement of synaptic inputs could contribute to nonlinear local integration. This may lead to computational capabilities such as directionally biased responses, proximal- distal synapse interplay, and classical RF–contextual interactions[62-64]. Intrinsic membrane properties[65] and inhibitory mechanisms[23,51,58,65-67] have also been proposed to explain multisensory responses of SC neurons. Future studies at single-neuron level[44,68-70] should explore how the functional arrangement of synaptic inputs contributes to multisensory integration, while research on long range connectivity should address the role of cortical input[60,71]. A genetic marker for identifying multisensory neurons would be a valuable tool for this purpose.

## Ethological relevance of audiovisual delay representation in the SC

The observed tuning of SC neurons to AV delays could have a clear ethological relevance. Auditory signals typically reach the SC ~50 ms earlier than visual signals[22,72] due to differences in receptor latencies—ranging from 20–100 ms in the retina[73] to 1–4 ms in the cochlea[74]. An optimal integration window for stimuli with a 50 ms asynchrony is consistent with the timing that evokes optimal perceptual coincidence[75,76]. However, SC neurons are tuned to a broad range of AV delays[23,24]. This broad tuning might support the binding of cross-modal stimuli despite differences in their arrival times, by providing a buffer for latency differences between sensory modalities[77]. The consistent difference in the speed of light and sound introduces a regular feature in multisensory signals: visual input consistently precedes auditory input in a distance-dependent manner, counterbalancing the different latency of the two sensory systems. Experience-dependent development of multisensory temporal windows aligns sensory integration with the statistics of natural events[76], and broad AV delay selectivity may enable robust detection across varying target distances. Our observation that nonlinear responses are overrepresented when visual signals precede auditory signals suggests that this mechanism might allow the efficient representation of the temporal features from the natural environment. It would also enable robust detection of external events even at low contrast visual input or low sound intensity which evoke neuronal responses with high latency variability. This adaptive strategy might ensure animals can respond effectively to dynamic and variable environmental stimuli. Analogous nonlinear mechanisms have been observed in the encoding of target distance through echo delay representation in bats[78,79], suggesting a conserved function across species for processing temporal disparities. Importantly, our approach to studying population coding in the SC relied on classification methods, including SVM and RFC. While the classifiers' ability to predict AV delays indicates that neural activity contains sufficient information to distinguish between stimuli, it does not reveal the underlying computational mechanisms. Moreover, multiple factors may contribute to the classifiers' performance. To address this, we conducted analyses with a range of controls aimed at disentangling the contribution of specific factors. These analyses emphasised the role of nonlinear interactions and the distinct contributions of specific subpopulations in representing AV delays. However, the extent to which this information is utilised by downstream circuits will ultimately determine the functional relevance of this representation.

## Non-uniform representation of AV delays in the SC

Our study revealed higher discriminability of AV delays in the medial-posterior SC, which encodes the upper temporal sensory field. Biased representation toward the temporal hemifield has been observed in SC-mediated behaviours, such as saccadic movement[80], covert orienting behaviour[81], unconscious priming[82] and saccadic latencies[83]. Biases along the anterior-posterior axis of the SC might be attributed to the non-homogenous representation of the visual field due to asymmetries in retinal projections[56,84-87]. Multisensory interactions

also exhibit an asymmetric expression in humans, where AV integration is stronger in the temporal hemifield than the nasal hemifield[48]. Eccentricity biases of multisensory interactions have also been observed in human cortical visual areas, where decoding of auditory information improves toward the peripheral retinotopic regions in early visual cortex (V1, V2, and V3)[88,89]. This eccentricity-based functional organisation is predicted by primate anatomical connectivity[90], where feedback projections from non-visual areas preferentially target the periphery of early visual areas. This differential multisensory representation may enhance reactions to peripheral, unattended stimuli—a critical function for survival.

In rodents, the medial SC is crucial for innate defensive responses to overhead looming stimuli[15] and plays a key role in escape behaviours[16]. Both the medial and posterior SC are involved in turning behaviour[12,91], achieved through a highly interconnected network with cortical and subcortical structures associated with visual and auditory processing, and visuomotor responses[28]. These connections support the SC's crucial role in guiding spatial navigation[92,93], visual attention[92,94] and orienting behaviour[12,95], which underscores the importance of achieving a precise representation of AV temporal discrepancies for accurately guiding animals in their behavioural responses. An accurate estimation of AV delays could serve as a mechanism for predator's distance estimation and therefore modulate the expression of defensive responses, such as arrest, freezing, or escaping[16,96,97]. To validate this, future work should address the behavioural relevance of using AV delays as a cue for the selection of defensive responses. Future investigations could disentangle the relative contribution of the spectral content of sound and light to AV delay representation across the visual field, i.e., focusing on high frequency sounds—which contribute to central RFs[98]—and ultraviolet (UV) light—overrepresented in the ventral retina[99,100]—and enhance our understanding of how this system supports behaviourally relevant computations in natural environments.

## Functional specificity of SC connectivity

Putative connectivity was assessed through cross-correlation analysis of inter-trial-interval responses across the SC's dorsoventral axis. Consistent with principles of connectivity found on the visual cortex[43,44], our results show that spatial RF correlation is a strong predictor of connectivity in the SC. Furthermore, connections within functional subclasses are overrepresented, with the most prevalent connections occurring between visual neurons. Interestingly, multisensory neurons receive approximately 50% of functional local inputs from multisensory neurons and the remaining from visual and auditory neurons, suggesting a complex interconnected network. Recurrent connectivity between multisensory neurons could extend the temporal window of integration[101] and explain the longer AV delay selectivity than the predicted by the linear summation of unisensory responses. The overall rate of connectivity decreased from the medial SC to the lateral SC, with delay-selective neurons displaying the lowest rate of connectivity in the lateral SC (Supplementary Fig. 4) consistent with lower AV delay encoding in lateral regions. Our jitter correction and high threshold over the noise distribution for estimating connectivity likely resulted in an underestimation of connection probabilities. Importantly, shared inputs were more uniformly distributed across SC regions, in contrast to the spatially structured patterns observed for putative local connections. This suggests a differential contribution of shared input and recurrent connectivity to multisensory integration across SC regions. Future work using multiple probes[102] should assess lateral connectivity potentially contributing to cross-modal interactions in the spatial domain. Furthermore, circuit mapping through viral tracing[103,104] or electrophysiological methods in slices[105] would be needed to confirm our results. However, previous work has shown that correlative analysis of neuronal responses measured in vivo can provide good estimates about synaptic connectivity measured in brain slices of the same tissue[106]. A highly interconnected network of multisensory neurons might provide a mechanism for extending the multisensory binding window, ensuring robust and precise encoding of sensory signals. This could ensure reliable detection of targets across a wide range of environmental conditions, while nonlinear integration enhances the precision of temporal information encoding.

## Methods

### Animals

All animal procedures performed in this study were licensed by the UK Home Office and approved by the Crick Institutional Animal Welfare Ethical Review Panel (PEB4A5081 & PP2817210). Experiments in this study were performed in 31 male and female C57BL/6 J.Cdh23[753A>G] mice (MRC Harwell, UK)[107], aged 2–4 months (4 females and 3 males for behavioural experiments and 13 females and 11 males for Neuropixels recordings). This mouse line has a CRISPR/Cas9- mediated homology repair of allele Cdh23 that prevents age-related hearing loss[107]. Mice were bred and maintained at the mouse facility of the Francis Crick Institute, with controlled temperature ($21 \pm 2\,°C$) and humidity ($55 \pm 10\%$ RH), with *ad libitum* access to food (Teklad global diet, Envigo, UK) and water, and kept on a 12:12 h light/dark cycle (lights on at 10 pm). All experiments were carried out in sound-attenuated chambers. Mice were randomly assigned to the different experimental groups. Sex-based analysis were not performed due to otherwise too small sample size.

### Surgical procedures

Mice underwent two surgeries (metal headplate implantation and craniotomy). Analgesia was provided on the day prior to surgery (Metacam 5 mg/kg in custard). For both surgeries, mice were anesthetised with isoflurane (5% induction, 2–2.5% maintenance) and injected subcutaneously with analgesic and anti-inflammatory compound (10 mg/kg Meloxicam and 0.1 mg/kg Buprenorphine, dosage of injectable 0.1 ml/10 g of body weight). The animals body temperature was maintained at 37 °C with a thermal mat. A metal headplate was fixed to the skull using dental cement (Super Bond C&B, Sun Medical) centred on the right SC (AP: −3.7 mm and ML: 0.9 mm from bregma). After up to 2 weeks of habituation to the behavioural apparatus, animals underwent a second surgery to perform a craniotomy of 2–3 mm in diameter and a durectomy. The craniotomy was covered with artificial dura (Dura-Gel, Cambridge NeuroTech) and sealed with Kwik-Cast silicone elastomer (World Precision Instruments). Mice were left to recover for at least 24 h before recordings.

### Neuropixels recordings

After recovery (minimum 24 h), neural activity was recorded using Neuropixels 1.0[29]. For 2–7 days Neuropixels recordings were performed on awake, head-restrained animals. Each animal underwent a maximum of 5 probe insertions, with no more than two insertions per day. The order of the anatomical location of the insertion was randomised. By varying the position of the recording site, neuronal activity was acquired along the medial-lateral axis (ML: 0–1.9 mm from midline) and anterior-posterior axis (AP: −3.08 to −4.72 mm from bregma) portion of the colliculus. Neuropixels recordings were performed using a National Instruments I/O PXIe-6341 module and OpenEphys software (https://github.com/open-ephys). The probe was coated with DiI (ThermoFisher Scientific, V22885) or DiO (ThermoFisher Scientific, V22886) on the day of the recording, to be able to validate recording location. On the last day of recordings, the animals were perfused and the extracted brains were imaged with serial two-photon tomography[108] to perform post-hoc tracing of the electrode locations. Brainreg was used to register the brains to a reference atlas[109] and the probes were traced with brainreg-segment[109].

During the recording, animals were allowed to run freely on a cylindrical treadmill made of polystyrene foam. The movement of the cylinder was recorded by a shaft encoder (US Digital, H5-100-NE-S). Additionally, in a subset of experiments, to record uninstructed body movements during sensory stimulation, videos of the face of the animals were acquired through a CMOS camera (Basler acA1440-220um), coupled with a 50 mm lens (Computar Lens M5018MP2). Videos were recorded at 200 Hz and the acquisition was synchronised with Neuropixels recording using a common TTL trigger sent by the HARP RGB LED board.

Sessions were automatically spike-sorted using Kilosort2 (https://github.com/MouseLand/Kilosort/releases/tag/v2.0) and manually curated to select isolated single cells using Phy (https://github.com/cortex-lab/phy). Units were classified as 'good' based on the following criteria: they exhibited a strong refractory period in the CCG, had a typical waveform, and maintained a stable FR throughout most of the recording. Recordings were excluded from further analysis if, upon visual inspection of the spike-sorted data and anatomical reconstruction of the probe track, the anatomical boundaries of the SC could not be determined. For the experiments using the AV delay protocol, a total of 106 recordings were performed on 24 animals. Of these, 14 recordings—obtained from 10 of these animals—were excluded from further analysis. The final dataset included 92 recordings from 24 animals. No recordings were excluded from the spatial RF analysis.

Sample size was estimated based on the expected effect size based on similar studies[14,22,33] and the current standard in mouse neuroscience studies.

## Stimuli

Mice were presented with visual, auditory, and multisensory stimuli generated using Psychophysics Toolbox (www.psychotoolbox.com) in Matlab. To present visual and auditory stimuli simultaneously at corresponding locations on the opposite side of the recorded SC, we designed a specialised device using 0.8 mm thick laser cut PCBs in collaboration with the *Making lab*, a technology platform from the Francis Crick Institute. The device we created consisted of a hemispheric array with seventy-five stimulation positions, each housing a speaker and an LED. These stimulation positions were located 18° apart in both azimuth (horizontal) and elevation (vertical) directions. As a result, the arrangement formed a spherical section spanning 180° in azimuth and 72° in elevation, with a radius measuring 18 cm. Details of the stimulation apparatus can be found at (https://github.com/FrancisCrickInstitute/Coliseum).

Visual stimuli were presented via a dimmable LED (2.4 lux, 4.3°, OSRAM SmartLED, LWL283-Q1R2-3K8L-1) and auditory stimuli consisting of white noise (1–22 kHz, 60–65 dB intensity) were presented via a speaker (VISATON, K 28 WPC BL). Light diffusers (1.5 cm diameter thick Formlabs clear resin V4, FLGPCL04) were positioned in front of each LEDs. The resulting power UV and green wavelength was $42.0 \pm 2.0 \, \mu W$ at 365 nm and $29.6 \pm 0.5 \, \mu W$ at 510 nm. For RF mapping visual and auditory stimuli were presented independently or simultaneously in 5 elevations (−36 to 36° with 18° steps) and 7 azimuths (between −162 and 0° with 18° steps), totalling 35 locations with 25 repetitions per stimulus.

To study the effect of AV temporal synchrony, we first determined the centre of the RF of the neurons in the contralateral side of the recording site, and then presented visual and auditory stimuli either alone or combined. When together, two combinations were tested: visual-leading and auditory-leading stimuli. In the visual-leading condition, the auditory stimulus was presented with a delay between 0 and 100 ms relative to the visual stimulus, with 10 ms increments. In the auditory-leading condition, the visual stimulus was delayed by 0 to 100 ms relative to the auditory stimulus, with 25 ms increments. In both protocols, stimulus duration was 100 ms and the number of repetitions was 50. Interspersed trials with no stimulation were used as within subject control. In all experiments the stimulus sequence was

randomised in a block design, all the stimulation conditions were randomised, and this was repeated the number of times previously specified per protocol.

## Data analysis

All data analysis was done in MATLAB 2022b or Python. Data analysis was performed blind to the anatomical location of the recording sites.

Three complementary datasets were collected.

1. Spatial RF Dataset: We recorded 979 neurons from 11 sessions in 9 mice using high-resolution mapping across 35 locations (5 elevations × 7 azimuths) with 25 repetitions per auditory, visual, and multisensory condition. This protocol provided prolonged recordings and high stimulus repetition, enabling robust RF estimates and connectivity analysis based on cross-correlation.
2. Visual-leading AV Tuning Dataset: We recorded 5360 neurons from 92 sessions in 24 mice during presentation of audiovisual stimuli with multiple temporal delays at a single spatial location. Coarse RF mapping preceded this protocol to centre stimuli within the centre of the population RF.
3. Visual-leading and Auditory-leading AV Tuning Dataset: We recorded 1104 neurons from 22 sessions in 6 animals.

## Classification of neurons for AV delay protocol

A total of 5360 neurons (92 recordings, 24 animals) were used for the analysis. Neurons were classified based on their response to different stimuli using the ZETA test[35]. To account for multiple comparisons, p-values were corrected using the Benjamini–Hochberg procedure for the number of stimuli. Neurons were categorised as visual-responsive if they showed a significant response solely to visual stimulation. Similarly, auditory neurons were defined as those that exhibited a significant response exclusively to auditory stimulation. Neurons were considered multisensory if their response pattern fell into one of the following categories: Bimodal—neurons that responded to both visual and auditory stimulation; Gated—neurons that exclusively responded to multisensory stimulation. Visual and auditory spatial RF overlap for individual neurons was not used as a selection criterion for the analysis of AV delay tuning.

## Spatial receptive field analysis

A total of 979 neurons (11 recordings, 9 animals) were used for this analysis. To estimate the RF of individual neurons, a 2-dimensional Gaussian was fitted to the peak FR response for each stimulus. We computed Pearson's correlation between the location of the centre of the mean visual RF and the mean auditory RF for individual AV neurons (n = 89 neurons) and for each recording (N = 11 recordings). Moreover, we analysed the spatial alignment of RFs at the population level (N = 11 recordings).

## Unisensory latency estimation

The spiking rate of individual unisensory neurons was binned in 1 ms bins during unisensory stimulation. The latency to peak for each neuron and condition was determined as the time needed to reach the maximum FR of the response profile. Onset and offset time windows were estimated based on the latency data. Gaussian kernel density estimation was applied separately to the distributions of latencies corresponding to the stimuli. This approach facilitated the identification of the underlying bimodal distribution of peak latencies, indicative of distinct response phases. The onset and offset windows for visual latencies where from 0 ms–128 ms to 129 ms–250 ms respectively. Onset and offset windows for auditory latencies where from 0 ms–76 ms to 77 ms–250 ms respectively.

## Delay tuning and selectivity indices

To assess if the AV delay multisensory conditions differed from the highest unisensory condition, we tested the null hypothesis that peak

FR to the multisensory and unisensory trials do not differ. To simulate data that would be observed under the null hypothesis, multisensory and unisensory trials were merged and then split randomly into two groups. We calculated the peak FR of such two groups by computing the mean FR rate over a 20 ms window centred around the maximum response time. The difference peak FR for the two groups was then computed. A bootstrap distribution of peak differences was then generated by 10,000 repetitions. The difference value obtained from the actual multisensory and unisensory groups was compared to the 2.5th and 97.5th percentiles of the bootstrap distribution. If the actual difference fell outside this range, it suggested that the null hypothesis could be rejected and that there was a significant difference between the conditions. After identifying the AV delay conditions that differed significantly from the highest unisensory condition, the preferred delay was determined as the one that elicited the maximum FR among the AV delays showing significant differences from the preferred unisensory modality.

The delay selectivity of the neurons was quantified using three measures: the specificity index, the FR modulation index and the RI.

The specificity index (SI) evaluates the proportion of stimuli that significantly exceed the unisensory stimulation over the total number of stimuli presented, as is quantified as follows:

$$SI = 1 - \frac{sig_{delays}}{N_{delays}} \qquad (1)$$

A SI value close to 1 indicates high specificity, with neurons responding uniquely to one AV delay condition. Conversely, a SI close to 0 suggests that the neuron exhibits low selectivity to individual delays, by responding to more AV delay stimuli.

The FR modulation index measures the change in FR between the preferred AV delay condition and the least effective AV delay. It was quantified as follows:

$$FR_{modulation\,index} = \frac{AV_{max} - AV_{min}}{AV_{max} + AV_{min}} \qquad (2)$$

Where $AV_{max}$ and $AV_{min}$ are the mean peak FR over a 20 ms window centred around the maximum response time for the preferred AV delay and the AV delay evoking the lowest FR, respectively. Higher values of this index indicate a greater change in FR between the preferred and least effective AV delays.

The RI corresponds to the mean Pearson's correlation coefficient for each trial's repetitions within the same trial type.

$$RI = \frac{2}{R(R-1)} \sum_{i=1}^{R-1} \sum_{j=i+1}^{R} r(x_i, x_j) \qquad (3)$$

where $R$ is the number of repetitions, $r(x_i, x_j)$ is the Pearson's correlation coefficient between the responses at the $i$th and $j$th repetitions and $x_i, x_j$ represent the response at repetition $i$ and $j$ respectively. Higher RI values indicate high reliability of neurons and low trial-to-trial variability. To determine the chance level reliability, the RI was calculated for randomly selected time intervals within the total spike train, where responses were sampled independently of the trial type.

## Multisensory interaction index

To measure the deviation of multisensory responses to the sum of unisensory responses a multisensory interaction index (MII) was computed as follows:

$$MII = \frac{AV - (A + V)}{(A + V)} \qquad (4)$$

Were AV constituting the peak FR of the responses to AV stimulation whereas A + V responses represent the peak FR of post hoc summed auditory and visually evoked unisensory responses. To generate A + V delay trials, the recorded auditory responses were shifted according to the delay of the auditory stimuli during the corresponding AV delay trials. The mean baseline response was subtracted to account for the shift in baseline in the summed responses. Nonlinear interactions were identified when there was a 50% change in FR between the observed AV response and the sum of unisensory responses. Positive MII values indicate supralinear multisensory interactions, whereas negative MII values indicate sublinear interactions.

## Spatial correlation analysis

To analyse the relationship between the neurons' spatial location and their MII, a LMM was used. The model included the animal ID of each neuron as a random effect with a varying intercept and a fixed slope. The MATLAB function '*fitlme*' was used, resulting in the following notation:

$$MII \sim 1 + ML_{position} + AP_{position} + DV_{position} + (1|animal\,ID) \qquad (5)$$

were $ML_{position}$, $AP_{position}$, and $DV_{position}$ correspond to the position of each neuron along the ML, anterior-posterior and DV axes, respectively.

## Population analysis

A linear classifier (SVM) and a nonlinear classifier (RFC) were used to estimate population delay decoding accuracy. All neurons were included in the decoder analysis. The spiking rate of individual neurons, binned in 10 ms bins during single trial stimulation, was used to decode the AV delay trials. Classifiers were trained with increasing number of neurons, reaching the maximum neuron's number in the population (5360) or the maximum neuron's number on individual recordings, when trained on single recordings.

Two main classifiers were trained. A first classifier was trained to estimate decoding accuracy between audio-visual trials and auditory shifted trials were included to account for the classification of delayed auditory responses. Auditory shifted trials were obtained by aligning the auditory response to the onset of auditory stimulation during AV delay trials.

All classifiers were trained on 80% of data and assessed on 20% held-out data. Since we always repeated the stimulus 50 times, this means that 40 trials were used for training and 10 trials were used for testing in all cases. We used a 5-fold cross validation and averaged the decoding accuracy of 5 runs. The decoding accuracy reported corresponds to the classification accuracy for AV delay trials only for the test set. The significance of the decoding accuracy (compared to chance) was computed by performing a Mann–Whitney U test comparing the observed classifier accuracy to the accuracy obtained when randomly permuting the labels of the trial types.

To assess decoding accuracy for subpopulations of neurons (mixed population, visual, auditory, AV delay-selective and delay-nonselective neurons), we fixed the number of neurons to control for decoding performance differences driven purely by population size (as shown in Fig. 4). To account for variability in neuron reliability and selectivity, we repeated the classification 20 times using resampling with replacement from the relevant subpopulation. This approach allowed us to examine the effect of diverse combinations of neurons on decoding performance. We chose this method because we hypothesise that population decoding is influenced not only by the number of neurons but also by their selectivity profiles. Thus, our approach emphasises the role of neuronal diversity in decoding. Linear classifiers were trained with 200 randomly selected neurons belonging to these subpopulations. Overall decoding accuracy, as well as misclassification rate, was estimated for 20 iterations of the classifier.

The misclassification rate was calculated as the mean false negative error for the specific misclassification between AV delay trials and auditory-shifted trials. For AV delay-selective and delay-nonselective multisensory neurons, a third classifier was trained to discriminate solely between AV delay trials. From this, the misclassification error (ms) was obtained by calculating the difference between the observed and predicted trial labels.

To characterise the decoding accuracy in different SC anatomical axes (ML, AP, DV), we varied the number of neurons used in the linear classifier. A subset of 25, 50, 75, or 100 neurons was randomly selected, and its accuracy was estimated over 20 iterations for this analysis. The final accuracy was calculated as the mean of these 20 repetitions. We also varied spatial bin size ranging from 250 to 650 μm in 50 μm steps. Decoding accuracy was also assessed by combining the anatomical axes. When the ML and AP axes were considered, the SC was divided into four anatomical bins (each 2.20 mm³, 1039 neurons, s.d. ± 121). When considering all three anatomical axes (ML, AP, DV), the SC was divided into eight anatomical bins (each 1.09 mm³, 528 neurons, s.d. ± 243). Finally, analysis on individual recordings accuracy was performed by randomly sampling 25 neurons from the available population, and the recording mean anatomical location was used. To analyse the relationship between classification performance and anatomical binning, a LMM was used. LMM analysis was used to investigate the relationship between classification accuracy along anatomical axes and the properties of the neurons used to train the decoders. This analysis utilised two sets of predictor variables. The first set comprised the proportions of visual, auditory, multisensory, and delay neurons in the training population. The second set consisted of neuron-specific properties: MII, absolute MII, FR modulation index, and RI. For each classification run, these variables were calculated for the neurons used to train the decoder. The distributions of these properties were then analysed, with kurtosis applied to evaluate their shape. If a distribution's kurtosis was below 2, it was considered symmetrical, and the mean was calculated. For distributions with a kurtosis above 2, indicating skewness, the median was used instead.

To assess the contribution of nonlinear interactions to decoding accuracy, we compared the performance of decoders trained on observed responses with those trained on synthetic linear sums. Both a linear classifier (SVM) and a nonlinear classifier (RFC) were used for this analysis.

## Connectivity analysis

Cross-correlation analysis was performed on spiking activity between 1 and 2 s after stimulus onset. The neuronal responses during this time window were concatenated across repetitions. For this analysis, we included all neurons that were responsive to any stimulus (visual, auditory, or multisensory), regardless of baseline FR. For non-responsive neurons, only those with a baseline FR of at least 10 Hz were included. This resulted in 3543 neurons being included, from 92 recordings and 24 mice. The CCG for a pair of neurons simultaneously recorded was calculated as described by Sielge et al.[39]:

$$CCG(\tau) = \frac{\frac{1}{M} \sum_{i=1}^{M} \sum_{t=1}^{N} x_1^i(t) x_2^i(t+\tau)}{\theta(\tau) \sqrt{\lambda_1 \lambda_2}} \quad (6)$$

Where $M$ is the number of trials, $N$ is the number of bins per trial, and $\tau$ is the time lag. The spike trains of the two neurons in trial $i$ are denoted by $x_1^i$ and $x_2^i$. The triangular function $\theta(\tau)$ corrects for the variation in the number of overlapping bins at different time lags, while $\lambda_1, \lambda_2$ are the mean FR of the two neurons, used to account for FR dependence. To correct for stimulus-locked correlations and slow temporal correlation, a jitter correction was applied, by subtracting to the original CCG a jittered CCG. A jitter window of 10 ms was used.

$$CCG_{corrected} = CCG_{original} - CCG_{jittered} \quad (7)$$

CCGs were considered significant if the peak occurred within a ± 5 ms window and exceeded five standard deviations above the mean of the noise distribution. The noise distribution was defined using the CCG values between 10 and 15 ms away from zero time. For each significant CCG, the peak lag was identified as the time difference between zero and the peak time. Significant correlations with a peak lag of 0 ms were excluded, as synchronous firing at 0 ms is typically attributed to shared inputs[110] rather than direct interactions. Therefore, only correlations with a peak lag between 1 and 5 ms were considered significant putative connections and included in further analyses.

The false positive rate of connections was estimated by performing CCG analysis on neurons sampled from different recordings, with the number of neurons considered matching those in each original recording. The depth distribution of the randomly selected neurons was also matched to that of the original neurons. This was achieved by dividing the SC depth into 13 equally spaced bins of 150 μm and selecting neurons according to these depth bins.

To investigate whether spatial RF similarity predicts connection probability, CCG analysis was additionally performed on a subset of data (11 recordings from 9 mice). Only sensory-responsive neurons were included in this analysis ($N = 246$ neurons). The same CCG computation and jitter correction procedure described above was applied. Spatial signal correlation between multisensory RFs was computed as the Pearson correlation coefficient between the spatial response profiles of each neuronal pair. Putative connections were defined as above. To assess whether functional connectivity increases with RF similarity, neuron pairs were binned by signal correlation, and connection probabilities were computed per bin. A Cochran–Armitage trend test was applied to evaluate monotonic changes in connectivity. To control for session-related dependencies, a null distribution was generated by shuffling signal correlation values within sessions (1000 iterations) while preserving the number of connections per bin, and recalculating the test statistic for each iteration.

## Sound-induced behavioural response analysis

For analysis on sound-induced behavioural responses, facial (PCs) were extracted using singular value decomposition as implemented in the Facemap toolbox[111] (https://github.com/MouseLand/facemap). The average z-scored motion energy weights for the first PC were compared across trial conditions for different animals. For decoding analysis, linear classifiers (SVM) were trained using either the first PC or the first 100 PCs to discriminate between different trial types.

## Statistical analysis

To test for statistical significance between groups where appropriate we used either paired or non-paired non-parametric tests (Wilcoxon signed-ranks test or Mann–Whitney U test). To assess delay tuning of neurons we used permutation test. To compare distributions, we used the Kolmogorov–Smirnov test. For analyses involving subpopulations in which neurons from different recordings were pooled, we employed a hierarchical bootstrap approach when applicable to account for the nested data structure (neurons within sessions, sessions within animals). In these cases, bootstrap samples were generated by first resampling animals with replacement, then resampling sessions within each animal, and finally resampling neurons within each session. For each sample, we calculated a summary statistic—such as a correlation coefficient or median difference—and derived a two-sided $p$-value from the empirical distribution as $P = 2 \min\{q, 1-q\}$, where q is the quantile of the observed statistic within the bootstrap distribution.

Statistical test details and $p$ values are provided in figures and/or legends.

## Study design

We used a within-subject design, in which individual animals were exposed to all trial types (auditory, visual, multisensory, and AV

delays). Due to the nature of experiment monitoring during electro-physiological recordings, the experimenter could not be blinded to condition. However, experimenters were blinded to condition during data processing, as all trial types were analysed together within each session and underwent the same processing pipeline. Importantly, data analysis was conducted blind to the anatomical location of the recording sites. Sample sizes were not predetermined but are consistent with those commonly used in the field[14,22,33] and with current standards in mouse neuroscience studies. For analyses involving sub-populations—where neurons from different recordings were pooled—we employed a hierarchical bootstrap approach, when applicable, to account for the nested data structure (neurons within sessions, sessions within animals). Where appropriate, LMMs were used, incorporating recording ID and mouse ID as random effects.

### Reporting summary

Further information on research design is available in the Nature Portfolio Reporting Summary linked to this article.

## Data availability

Pre-processed data (spike sorted data) have been deposited in Figshare under accession code https://doi.org/10.25418/crick.28685360.v1, ref. 112. Raw electrophysiological data has not been deposited due to size and similarity with pre-processed data but are available from the authors upon request. The open-source designs of the speakers and LED device and its associated circuits is available through our institute scientific hardware platform (https://github.com/Iacaruso-lab/Coliseum).

## Code availability

Matlab and python code for analysing the data and generating the figures is available at https://doi.org/10.5281/zenodo.16901292, ref. 113.

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

## Acknowledgements

We would like to thank Tom P.A. Warner, Cecilia Della Casa, Andreas Schaefer and members of the Crick Neurophysiology community for useful discussions and feedback on the work, and Alexander Egea Weiss, Daniel R. Ward and Sadra Sadeh for comments on early versions of the manuscript. The work was supported by the Engineering and Physical Sciences Research Council (BBSRC, award ref. EP/X020924/1, M.F.I.) and the Francis Crick Institute which receives its core funding from Cancer Research UK (10746, CC2118, M.F.I.), the UK Medical Research Council (10746, CC2118, M.F.I.), and the Wellcome Trust (10746, CC2118, M.F.I.).

## Author contributions

M.F.I. and G.B. conceptualised the project. G.B. curated the data, carried out formal analysis, conducted the investigation G.K., X.C.F., and A.I. developed the stimulation apparatus. I.R. conducted and designed pilot experiments with the supervision of M.J.M. and M.F.I. secured funding and provided supervision. M.F.I. and G.B. wrote the original draft. All authors reviewed and edited the manuscript.

## Funding

## Competing interests

The authors declare no competing interests.
