## [Transparent Peer Review file · Nature Communications]

Functional specialisation of multisensory temporal integration in the mouse superior colliculus

Corresponding Author: Dr M. Florencia Iacaruso

Version 0:

Reviewer comments:

Reviewer #1

(Remarks to the Author)

In this manuscript, Bianchini et al. investigate the temporal integration of visual and auditory signals in the mouse superior colliculus (SC), examining responses at the levels of single neurons and neuronal populations. The authors employ cutting-edge extracellular electrophysiology, yielding extensive neuronal data and providing valuable insights into multisensory integration across the SC. However, it would be helpful to clarify how this dataset compares to previous work, particularly Ito et al. (2021).

The manuscript thoroughly characterizes the spatial distribution of visual, auditory, and multisensory responses elicited by flashes of light and bursts of white noise, confirming and extending prior findings. Notably, the authors identify a distinct neuronal subpopulation selective for temporal mismatches between visual and auditory inputs and demonstrate the significance of multisensory nonlinear response summation in encoding temporal asynchrony. Furthermore, they reveal an anisotropic encoding accuracy across the SC, highlighting enhanced representation in the medial-posterior region associated with the upper temporal visual field. Additionally, the study provides evidence of strong dorso-ventral interconnectivity among neurons sharing similar functional properties.

Overall, the data presented are of high quality, obtained through rigorous and appropriate methodologies, carefully analyzed, clearly interpreted, and sufficiently detailed to support the manuscript's conclusions. These findings advance our understanding of the functional organization of the SC and the mechanisms underlying the encoding of multisensory temporal asynchronies at both single-neuron and population levels.

This manuscript is suitable for publication in Nature Communications and is nearly ready in its current form. Below, I provide several suggestions aimed at further enhancing the clarity and impact of the manuscript.

MINOR CONCERNS

1. Page 3: The authors mention testing "from a range of spatial locations." Please specify this range explicitly in terms of azimuth and elevation.
2. Figure 1: Was retinotopy confirmed by correlating anatomical locations with receptive field locations? Including a supplementary figure to demonstrate this confirmation as proof-of-principle would strengthen confidence in the methods.
3. Page 3: "white noise (1-22Hz)." Considering the mouse auditory range extends up to ~100 kHz, discuss potential impacts of testing only a limited auditory frequency range. Similarly, specify the LED power at different wavelengths and discuss potential implications of retinal functional specialization along the green-UV and upper-lower visual field axes on the observed results.
4. Figure 1D: Were any notable spatial patterns observed in the distribution of visual, auditory, and multisensory responses across the SC along different axes? Consider adding a supplementary density map (similar to Suppl. Fig. 4a) to visualize subtle patterns more clearly, especially in relation to the medial-posterior enhanced representation of asynchrony encoding.
5. Figure 1D-right: It would improve clarity to overlay distributions directly to facilitate comparison. Additionally, could a more

precise analysis be performed to check for specific layer preferences beyond mere depth?

6. Figure 1H/L: Single-neuron audio-visual RF locations appear more correlated along azimuth than elevation, an intriguing observation. Could authors discuss whether this could result from elongated or less defined receptive fields along the dorso-ventral axis?
7. Figure 2F (Pages 5-6): The authors note "the mean preferred AV delay was 54 ms." Given the relatively flat distribution of AV delays and the tested range (0-100 ms, mean ~50 ms), clarify if the observed mean is genuine or possibly influenced by experimental design. Consider rephrasing the sentence on page 6 to reflect this clearly.
8. Page 20, first line: Clarify if the "mean FR over a 20 ms window" refers specifically to a window centered around the peak response.
9. Citation 23 (Miller et al., 2015): The authors found "Larger multisensory responses when stronger responses were advanced relative to weaker responses." Was this observed in the current dataset as well?
10. Figure 2L: Add a visual indication of the mean (e.g., a small downward-pointing triangle) to the plot for clarity.
11. Page 21 (CCG analysis): The authors mention, "only neurons with a baseline FR of at least 10 Hz were included." Given supplementary Fig. 5B suggests this threshold should exclude more than half of the neurons, clarify how 3543 neurons were included.
12. Page 9, Fig. 3C: Clarify the connectivity rate discrepancy—0.9% or 1.39%?
13. Page 9: "Multisensory neurons were more strongly connected with other multisensory neurons." However, Fig. 3D indicates different subtypes (D and M-D) have stronger connections to visual (V) neurons. Clarify or rephrase to state that multisensory neurons preferentially connect within their own subtype (D or M-D).
14. Figure 3F: Maintain consistency in the ordering of neuron subtypes throughout figures and within individual panels for clarity.
15. Figure 3G: Enhance the differences in line thicknesses in the schematic to make variations in connectivity clearer and easier to perceive.
16. Page 15: Correct the typo "genetic market" to "genetic marker."
17. Page 18: Correct or update the Coliseum link (<https://github.com/FrancisCrickInstitute/Coliseum>) as it is currently nonfunctional.

Reviewer #2

(Remarks to the Author)

Bianchini et al. present the functional specialization of multisensory (audiovisual) temporal integration in the mouse superior colliculus (SC). Using electrophysiological recording with a Neuropixel probe, they recorded over 5,000 neurons across anatomical axes of the SC, demonstrating how the spatially and functionally distinct SC neurons represent the audiovisual delay. The authors provide new insights into how the SC encodes audiovisual timing, particularly emphasizing the role of nonlinear integration and regional connectivity in supporting precise temporal representation of audiovisual inputs.

More specifically, they performed analyses from single neurons to populational levels and found how the audiovisual temporal delay is differentially encoded in AP, ML, and DV locations in the SC. The data strongly support the following: 1) non-linear summation in the SC is important for decoding audiovisual temporal delay; 2) higher recurrent connectivity (more functionally connected neuronal pairs) in the medial SC; 3) in correlation with this, robust encoding of audiovisual temporal delay in the posteromedial SC.

The experimental approach is technically impressive, and the authors present a compelling case for the role of nonlinear integration and regional connectivity in supporting precise temporal representations of multisensory inputs. However, several key concerns were raised, and we hope the authors can address those by conducting additional experiments, analyzing, and clarifying or toning down interpretations to enhance the scientific clarity of the conclusion and consolidate their story.

1. Additional experiments

1) Causality of recurrent connectivity

While the authors observed strong correlations between encoding of audiovisual temporal delay and putative functional (synaptic) connections measured by CCG, they did not provide direct causal evidence. Furthermore, their CCG analysis is still correlative to synaptic connectivity. Additional validation (e.g., optogenetic perturbations or in vitro paired recordings) would strengthen causal claims about recurrent circuits shaping multisensory integration.

Furthermore, the authors excluded lag=0ms peaks from the CCG to avoid shared input artifacts. However, shared inputs

might be segregated across SC areas and more strongly affect audiovisual integration differentially across SC regions (compared to recurrent connections). This potential must be tested or discussed further.

2) No behavioral correlates

The authors proposed that the decoding accuracy of the audiovisual temporal delay in the posteromedial SC is inferred from physiology alone. To clearly support their idea, they must test behavioral correlates (e.g., reaction time or detection thresholds for AV delays) while disrupting activity correlates in the SC. The behavioral experiment can also support their ecological interpretation that visual-leading AV stimuli evoking stronger supralinear integration are linked to natural statistics of light and sound propagation.

2. Additional analysis

1) Populational receptive field topography according to SC coordinate.

The authors propose that the posteromedial SC is specialized for encoding AV delays in the peripheral sensory space. While prior work they mentioned supports this, the current manuscript lacks direct analysis linking SC coordinates to receptive field eccentricity. Spatial mapping of RF eccentricity or azimuthal bias plotted along mediolateral and anterior-posterior coordinates will strengthen the authors' claim.

2) Contribution of supralinear vs. sublinear multisensory neurons to audiovisual delay encoding.

The identification of supralinear and sublinear multisensory neurons is an important observation. However, the manuscript does not explore how these subtypes contribute to AV delay encoding. A comparison of decoder accuracy or classifier weight between these groups, or even their distribution across regions, would enhance understanding of how nonlinear integration shapes temporal encoding.

3) Correlation between connectivity strength and receptive map overlap

The cross-correlation analysis yields putative functional connections, but it is unclear whether these connections occur preferentially between neurons with similar spatial tuning. Investigating the correlation between connection strength and the degree of receptive field overlap would provide insight into how spatial and temporal integration are jointly organized in SC microcircuits.

4) Visual neuron proportion and decoder performance (Figure 4)

There is an apparent discrepancy between poor decoding performance by visual neurons alone (Figure 4E, G) and the positive correlation between decoder accuracy and visual neuron proportion in mixed populations (Figure 4I). A discussion or control analysis explaining how visual neurons may contribute indirectly to decoding (e.g., by stabilizing baseline population activity) would be helpful.

3. Clarifying methods and data interpretation

1) Decoding accuracy vs. subpopulation distribution (Figure 6C)

While the authors conclude there is no significant correlation between decoding accuracy and neuronal subpopulation distribution across SC regions, several P values, particularly for auditory neurons, are relatively low ($P=0.06$ for auditory neurons; $P=0.1$ for multisensory neurons). It would be more accurate to interpret these as trends rather than clear null results. Toning down this conclusion would avoid overstatement.

2) Spatially misaligned receptive fields (RFs)

All AV delay experiments were conducted using spatially coherent audiovisual stimuli, yet Figure 1H and 1L show that individual neurons' RFs are not perfectly aligned across modalities. It is unclear how such misalignment was handled in decoding and summation analysis. Did the authors exclude or control for neurons with divergent auditory and visual RFs? A more detailed description of this criterion is essential to validate the population analysis.

3) Statistical significance in Figure 2L

The manuscript claims that observed preferred AV delays are longer than those predicted by linear summation. However, the reported mean difference (~13ms) appears modest, and no statistical test is provided. To support this conclusion, a formal statistical comparison (e.g., paired t-test or Wilcoxon signed-rank) should be included.

4) Decoder generalization with few neurons

The finding that only 9 neurons are sufficient to decode AV delay with above-chance accuracy is intriguing. However, more detail is needed on whether these neurons were sampled across multiple SC regions or within a single anatomical subdomain. This clarification would help interpret the generalization of local population coding.

5) Minor comments

- Mismatch in data values

We found some numbers are mismatched between the manuscript and the figures.

1) [Figure 1C] The proportions of visual, auditory, and multisensory neurons do not match (13%, 9%, and 34% in the manuscript vs. 16%, 14%, and 26% in Figure 1C).

2) [Figure 3C] "The percentage of connections between visual neurons was 6.17%, while auditory neurons received the highest percentage of connections from other auditory neurons, albeit their connectivity rate was much lower (0.9%)" If the latter is about the auditory-to-auditory connection, 0.9% should be 1.39%.

The authors should carefully review the manuscript to ensure all numbers are correctly written.

- A typo in the Discussion:

“A genetic marker for identifying multisensory neurons would be a valuable tool for this purpose.”: ‘Market’ to ‘marker’

Reviewer #3

(Remarks to the Author)

In this study, the authors propose the existence of distinct functional neuronal subtypes in the superior colliculus based on their responsiveness to different stimuli. One of these subtypes appears to be specifically tuned to asynchronous combinations of visual and auditory stimuli, enabling the encoding of temporal delays between them. Using machine learning techniques, specifically support vector machine (SVM) classifiers, the authors suggest that a subset of these neurons contains sufficient information to encode stimulus delay. This leads them to propose that the superior colliculus may employ a population coding mechanism to represent temporal delays. Their results indicate that a significant portion of the information used by the classifiers to predict delay is linked to the nonlinear summation occurring within this multisensory neuronal subtype. Moreover, they demonstrate that the capacity to encode this information is not homogeneously distributed across the superior colliculus but is particularly enhanced in its posteromedial regions. This region is associated with the upper-temporal visual field, prompting the authors to speculate that encoding delays between visual and auditory stimuli may contribute to threat detection and predator avoidance in animals.

The experiments presented in this study appear to be meticulously conducted, with a strong emphasis on data quality and methodological rigor. To my knowledge, the authors employ state-of-the-art extracellular recording technology, specifically Neuropixels probes, and utilize an automated spike-sorting method that is subsequently reviewed by human experts to exclude low-quality units. Additionally, they complement their electrophysiological recordings with video tracking of animal movements to account for potential motor influences on neuronal responses. For electrode localization, they implement an advanced approach that surpasses traditional histological techniques. A robust quality control process is also in place to exclude recordings that fail to meet predefined anatomical and physiological criteria. I commend the authors on their rigorous experimental approach.

However, I have a few concerns regarding methodological transparency and statistical analyses. Firstly, the authors do not explicitly state whether any animals or recordings were excluded from the study. While they mention performing an a priori sample size estimation, details regarding these calculations are not provided, even in the supplementary materials. Including this information would improve the study's ethical transparency and serve as a valuable example of best practices for the scientific community.

Second, the study employs a clearly hierarchical dataset: neurons are nested within recordings, and recordings are nested within animals. This structure implies that neurons cannot be considered independent observations, which has important implications for statistical analyses. The authors correctly use linear mixed-effects models in some parts of the study to account for these dependencies, demonstrating awareness of the issue. However, it is unclear whether this approach is consistently applied. In particular, the use of the Kruskal-Wallis test to compare neurons raises concerns, as this test assumes independent observations. While this may not substantially alter the study's main conclusions, it increases the risk of pseudoreplication, potentially leading to inflated statistical significance and a higher likelihood of Type I errors. A thorough review of the statistical tests used would strengthen the study's methodological rigor.

A third point concerns the reporting of SVM results. Confidence in the analysis would be enhanced if the authors reported both training and test accuracy, allowing readers to better assess model performance and potential overfitting. Additionally, it is unclear whether the confusion matrices presented in the figures correspond to the training or test set. Clarifying this aspect would improve the interpretability of the findings. Furthermore, while the methods section describes the dataset split (80% training, 20% testing with k-fold cross-validation) for the full dataset (5,360 neurons), it remains uncertain whether this approach is consistently applied across all SVM implementations, particularly when working with smaller subsets. The authors mention performing 20 iterations of classification in these cases, but they do not specify whether the selected neurons are randomized or remain the same across iterations. This is not a standard approach in machine learning, at least as commonly practiced, and raises concerns about its validity. Clarifying this methodological choice and providing a rationale for it would improve the study's robustness.

Overall, the manuscript is well written and clearly structured. The introduction is well-articulated and supported by appropriate references. While multisensory integration is outside my primary area of expertise, from the perspective of a researcher interested in developmental neurobiology and circuit formation, I find the authors' effort to quantify neuronal subtype connectivity within a circuit model particularly valuable. Their findings contribute new insights into collicular circuitry in adults, providing a reference for future studies investigating superior colliculus development.

Based on an initial review of the cited literature, my impression is that the study primarily contributes to confirmatory science, reinforcing existing findings on receptive field alignment across sensory modalities, collicular neurons tuned to visual-auditory delays, nonlinear summation in multisensory neurons, and the functional specialization of different collicular regions.

Regarding the study's conclusions, I suggest refining the strength of claims regarding population coding based on SVM classifier results. The ability of a classifier to predict stimulus delays with high precision does not imply causality or provide direct biological insight into how the information is represented in the circuit. SVM is a statistical tool designed to identify patterns in data; its success indicates that neural activity contains sufficient information to distinguish between stimuli but does not reveal the underlying computational mechanism. Multiple factors could contribute to the classifier's performance.

Addressing this limitation in the discussion—along with clearer definitions of concepts such as distributed and sparse coding—would strengthen the interpretation of the results.

In conclusion, this is a well-executed study that merits publication. However, I recommend that the authors address the concerns raised here, particularly regarding statistical analyses, methodological clarity, and the interpretation of SVM results.

As a minor issue, I suggest reviewing figure labeling and references in the text. For instance, the labels for panels G, H, I, and J in Figure 2 appear to be incorrect.

Reviewer #4

(Remarks to the Author)

Version 1:

Reviewer comments:

Reviewer #1

(Remarks to the Author)

The authors have addressed all our concerns.

Reviewer #2

(Remarks to the Author)

The authors have addressed all concerns we raised in the revised manuscript. We have only two minor comments.

1. In their response regarding the new analysis on RF similarity and connection probability, the authors mention that this was conducted in a separate dataset, stating “This has been performed in a separate dataset and shows results similar to those previously found in the visual cortex ...”. It would be helpful if the authors could clarify why the original dataset was not used for this analysis. Additionally, we recommend including details on the mice used, the number of units per recording, and the analysis performed on each group in the “Source Data” file.

2. In the discussion section, the authors propose that audiovisual delay may influence the decision between freezing and escaping. However, most previous studies on freezing and escape behavior have used looming shadow stimuli, whereas the current study uses a flash. Liang et al. (2015) also note that flash-evoked arrest during locomotion is behaviorally distinct from freezing. We encourage the authors to either provide additional references supporting flash-induced freezing behavior or elaborate further on how a visual flash could trigger both freezing and escape responses.

Reviewer #3

(Remarks to the Author)

The authors have adequately addressed the reviewers' previous comments, resulting in a significantly improved version of the manuscript. I have only identified a few very minor issues in this second review:

Page 7, first paragraph (end): The figure reference appears to be incorrect. It currently points to Supplementary Figure 2, but it likely should refer to Figure 1G.

Figure 2J: The p-value displayed in the figure has not been updated to match the one reported in the main text (page 8, paragraph beginning “Given the similarity of the...”). A similar discrepancy is observed in Supplementary Figure 3, where the p-values shown in the figure do not match those in the text. This issue may affect other supplementary figures as well. I recommend a thorough consistency check. Additionally, the figure legends for Figure 2 and Supplementary Figure 3 still refer to the previous statistical method (“linear regression”) instead of the updated one (“LMM”).

Page 8, final paragraph (“Linear summation characterised...”): When reporting the MII values for conditions in which the leading stimulus is auditory or visual, the median is cited as the central tendency measure. However, no measure of variability is provided. Based on the corresponding boxplots, it appears that the interquartile range (IQR) is used and should be explicitly mentioned.

Supplementary Figure 6: The statistical test applied does not appear to have been updated to address pseudoreplication, as requested in the previous review round.

Page 13, paragraph 2 (final sentence): The sentence “However, connectivity between delay-selective neurons was not

explained by their preferred delay similarity (Suppl. Figure 5)” seems to cite the incorrect supplementary figure.

Page 18, beginning of the paragraph (“Previous studies have shown asymmetries...”): Reference [46] (Bertini et al., 2008) is formatted inconsistently compared to the other citations.

Reviewer #4

(Remarks to the Author)

REVIEWER COMMENTS

Below, we provide a detailed reply to the reviewers' comments. We thank the reviewers for their time and constructive feedback. We believe that addressing these comments has significantly improved the manuscript.

The most substantial revisions include:

1. **Analysis of connection probability as a function of spatial receptive field similarity.** This analysis, conducted on an independent dataset, revealed results consistent with findings in the visual cortex, where spatial receptive field correlation is a strong predictor of connection probability (Cossell et al., Nature, 2015; Iacaruso et al., Nature, 2017; The MICrONS Consortium, Nature, 2025). Our findings extend this principle to the SC and support our connectivity model linking different sensory modalities.
2. **Validation of receptive field (RF) location as a function of anatomical position within the SC.** This analysis was requested by reviewers 1 and 2 to confirm known topographic organisation and strengthen confidence in our methodology.
3. **Analysis of zero-lag correlations to investigate shared input.** This enabled us to differentiate the contributions of shared inputs and recurrent connectivity to multisensory integration across SC regions.
4. **Implementation of hierarchical bootstrap methods.** In analyses involving pooled neuronal subpopulations, we employed hierarchical bootstrapping to account for the nested structure of our data (neurons within sessions, sessions within animals), as recommended by reviewer 3.

Additional changes include: expanded analysis of the spatial distribution of neuronal subpopulations, improvements to the clarity and detail of the Methods section, and responses to all remaining reviewer questions and requests for clarification. In addition to the requested changes, as detailed below, we have also adjusted the sign of the azimuth axis in the receptive field (RF) mapping shown in Figure 1. This modification does not affect the results in any way.

Reviewer #1 (Remarks to the Author):

In this manuscript, Bianchini et al. investigate the temporal integration of visual and auditory signals in the mouse superior colliculus (SC), examining responses at the levels of single neurons and neuronal populations. The authors employ cutting-edge extracellular electrophysiology, yielding extensive neuronal data and providing valuable insights into multisensory integration across the SC. However, it would be helpful to clarify how this dataset compares to previous work, particularly Ito et al. (2021).

The reviewer is correct that it is pertinent to compare our results with those of Ito et al., 2021¹. We have now been more explicit in the discussion.

Our findings are generally consistent with those previously reported by Ito et al.¹ who showed that the mouse SC contains topographically organized visual and auditory neurons that exhibit nonlinear multisensory integration. Their model-based analysis, based on the neuronal responses to a fixed AV delay, showed that multisensory integration is driven by an early auditory response modulated by a subsequent visual response. Our approach enabled the characterisation of AV integration in the temporal domain using a range of temporally shifted auditory and visual stimuli, which evoked stronger nonlinear integration when visual stimuli preceded auditory. We identified AV delay-selective neurons, a subset of bimodal and gated multisensory neurons, that exhibit nonlinear summation of sensory inputs at specific temporal disparities. Our results show that unimodal response latency is a

poor predictor of audio visual delay selectivity, highlighting the relevance of exploring the multidimensional parameter space.

The manuscript thoroughly characterizes the spatial distribution of visual, auditory, and multisensory responses elicited by flashes of light and bursts of white noise, confirming and extending prior findings. Notably, the authors identify a distinct neuronal subpopulation selective for temporal mismatches between visual and auditory inputs and demonstrate the significance of multisensory nonlinear response summation in encoding temporal asynchrony. Furthermore, they reveal an anisotropic encoding accuracy across the SC, highlighting enhanced representation in the medial-posterior region associated with the upper temporal visual field. Additionally, the study provides evidence of strong dorso-ventral interconnectivity among neurons sharing similar functional properties.

Overall, the data presented are of high quality, obtained through rigorous and appropriate methodologies, carefully analyzed, clearly interpreted, and sufficiently detailed to support the manuscript's conclusions. These findings advance our understanding of the functional organization of the SC and the mechanisms underlying the encoding of multisensory temporal asynchronies at both single-neuron and population levels.

This manuscript is suitable for publication in Nature Communications and is nearly ready in its current form. Below, I provide several suggestions aimed at further enhancing the clarity and impact of the manuscript.

We thank the referee for the valuable review, the careful consideration of our manuscript and the detailed suggestions below. We hope we have addressed all the concerns and in doing so we have enhanced the clarity of the manuscript.

MINOR CONCERNS

1. Page 3: The authors mention testing "from a range of spatial locations." Please specify this range explicitly in terms of azimuth and elevation.

Details were provided in methods but we have now also included them in the results:

We used a custom-made hemispheric stimulation apparatus to provide spatially coincident visual and auditory stimuli from a range of spatial locations (Figure 1B), spanning 162° in azimuth and 72° in elevation across recordings.

2. Figure 1: Was retinotopy confirmed by correlating anatomical locations with receptive field locations? Including a supplementary figure to demonstrate this confirmation as proof-of-principle would strengthen confidence in the methods.

This was a very good suggestion, and we have now included a plot of receptive field location as a function of anatomical location in Supplementary Figure 2. We plotted the auditory and visual RF positions against anatomical location for the experiments in which detailed RF mapping was performed (corresponding to the data in Figure 1).

We find that both visually and auditory responsive neurons exhibit a strong correlation between their anatomical location along the anterior-posterior axis and their RF location (Supplementary Figure 2A). Specifically, the visual map has a $74 \pm 25^\circ/\text{mm}$ slope and $30 \pm 23^\circ$ offset, while the auditory map has a $73 \pm 22^\circ/\text{mm}$ slope and $43 \pm 20^\circ$ offset. In contrast, no such correlation was observed for elevation along the medial-lateral axis, consistent with previous results².

3. Page 3: "white noise (1-22Hz)." Considering the mouse auditory range extends up to ~100 kHz, discuss potential impacts of testing only a limited auditory frequency range. Similarly, specify the LED power at different wavelengths and discuss potential implications of retinal functional specialization along the green-UV and upper-lower visual field axes on the observed results.

To measure auditory receptive fields, we used white noise of 1-22kHz covering the peak of the hearing range of mice³. However, neurons with frontal RFs have been shown to represent spectral cues to encode azimuthal auditory space². Since frequencies in the range of 10–80 kHz potentially contribute to frontal receptive fields, our stimulus should capture asymmetries between the contralateral and ipsilateral ear already present in frequencies ranging from 10-22kHz, this is supported by Figure 1 showing that the white noise used in our study allowed us to capture neurons with frontal receptive fields, but we have now addressed in the discussion the potential lower proportion of frontal auditory receptive fields captured with our stimuli.

A description on the spectrum of the white LEDs has now been added to the manuscript's methods. Given that the intensity of our white LEDs at UV wavelength is very low, we expect that mainly M cones and rhodopsin are activated in our system. Previous work has shown an asymmetry in the sensitivity for UV light in the dorso-ventral axes of the retina. We have now added in the discussion that the visual responses in our study are mainly driven by green sensitive photoreceptors given the light emission characteristics of the LEDs used in the study (page 19, Methods section). Future work should address how a more varied stimulation protocol might capture temporal encoding of audiovisual stimuli with different spectral properties present in the natural environment.

Specific change in the methods: *Visual stimuli were presented via a dimmable LED (2.4 lux, 4.3°, OSRAM SmartLED, LWL283-Q1R2-3K8L-1) and auditory stimuli consisting of white noise (1-22 kHz, 60-65 dB intensity) were presented via a speaker (VISATON, K 28 WPC BL). Light diffusers (1.5 cm diameter thick Formlabs clear resin V4, FLGPCL04) were positioned in front of each LEDs. The resulting power UV and green wavelength was $42.0 \pm 2.0 \mu\text{W}$ at 365 nm and $29.6 \pm 0.5 \mu\text{W}$ at 510 nm.*

Specific change to the discussion: *An accurate estimation of AV delays could serve as a mechanism for predator's distance estimation and therefore guide the rapid selection of defensive responses, such as freezing or escaping. To validate this, future work should address the behavioural relevance of using AV delays as a cue for the selection of defensive responses. Future investigations could disentangle the relative contribution of the spectral content of sound and light to AV delay representation across the visual field, i.e. focusing on high frequency sounds—which contribute to central receptive fields² and ultraviolet (UV) light—overrepresented in the ventral retina^{4,5}—and enhance our understanding of how this system supports behaviorally relevant computations in natural environments.*

4. Figure 1D: Were any notable spatial patterns observed in the distribution of visual, auditory, and multisensory responses across the SC along different axes? Consider adding a supplementary density map (similar to Suppl. Fig. 4a) to visualize subtle patterns more clearly, especially in relation to the medial-posterior enhanced representation of asynchrony encoding.

Thanks for the suggestion. We have added Supplementary Figure 2 showing the density map of visual, auditory and multisensory neurons across the different regions of the SC. Please also consider Figure 6c in which we have investigated the proportion of difference functional cell types in the different anatomical regions of the SC, and as reported in the manuscript *our analysis revealed no significant correlation between decoding accuracy and the distribution of subpopulations (for percentage of visual neurons: $P = 0.4$, auditory: $P = 0.06$, delay: $P = 0.1$, multisensory delay-nonselective neurons: $P = 0.1$,*

LMM, **Figure 6C**). We believe that the differential distribution of these cell types contributes to differences in population encoding of audiovisual delay. But these differences could not be specifically attributed to any specific cell type.

5. **Figure 1D-right**: It would improve clarity to overlay distributions directly to facilitate comparison. Additionally, could a more precise analysis be performed to check for specific layer preferences beyond mere depth?

We thank the reviewer for this suggestion. To improve clarity, we have now added an overlay of the distributions to facilitate comparison in Figure 1C.

In response to the request for more precise layer analysis, we attempted to further resolve sublayer distinctions within the SC. Our current method combines electrophysiological features and histological reconstruction of fluorescently-labelled probe tracks. While this approach allows us to delineate approximate SC boundaries, the resolution and variability along anatomical axes limit our ability to define individual sublayers.

To further investigate potential layer-specific differences, we explored the use of LFP analysis, which has been shown to help differentiate cortical layers^{6,7}. We applied the IBL LFP analysis pipeline (<https://github.com/int-brain-lab/ibllib>) to both our dataset and a publicly available IBL Neuropixels recording that spans retrosplenial cortex (RSP) and the SC.

However, we found that LFP features were not consistent enough to support sublayer identification in the SC. Specifically, as shown in the Figure, the power spectrum did not show clear transitions at the RSP–SC boundary, and spectral patterns varied substantially across probe insertion sites. This spatial variability and lack of reproducible laminar signatures suggest that, unlike in cortex, LFPs do not reliably reflect layer structure in the SC.

Comparison of LFP power spectra across depth in a representative Neuropixels recording from our dataset (left) and a corresponding IBL dataset (right). Each column shows the LFP power as a function of depth, aligned to anatomical regions identified through histological reconstruction. While laminar transitions are evident in cortical regions, there is no clear spectral discontinuity at the RSP–SC boundary in either dataset. Moreover, variability in the spectral pattern across insertion locations complicates the use of LFP features for reliable SC sublayer identification.

6. Figure 1H/L: Single-neuron audio-visual RF locations appear more correlated along azimuth than elevation, an intriguing observation. Could authors discuss whether this could result from elongated or less defined receptive fields along the dorso-ventral axis?

The referee is correct in his observation that audio visual receptive fields are more correlated in azimuth than elevation. Consistent with our results, Ito et al., 2020² found “a strong correlation between the A–P position of the auditory neurons in the SC and their RF azimuths (Fig. 2e, $r = 0.70$). The correlation of the M–L neuron position and RF elevation was weaker ($r = 0.22$). This is consistent with a previous report that showed that mice can discriminate two sound sources better along the azimuthal axis than the elevation axis (azimuthal: $31 \pm 6^\circ$; elevation: $80.7 \pm 1.7^\circ$)⁸”. We agree with the reviewer that this could be explained by more elongated RFs in elevation. Unfortunately, we believe the separation of our speakers (18 degrees) does not give us enough resolution for a precise comparison of the width and length of the receptive fields along azimuth and elevation.

7. Figure 2F (Pages 5-6): The authors note “the mean preferred AV delay was 54 ms.” Given the relatively flat distribution of AV delays and the tested range (0-100 ms, mean ~50 ms), clarify if the observed mean is genuine or possibly influenced by experimental design. Consider rephrasing the sentence on page 6 to reflect this clearly.

Thank you for the suggestion. We agree that the observed mean could reflect the centre of the tested range rather than a true preference. As suggested, we have clarified the wording to avoid overinterpretation.

Specific change to the text: Consistent with previous reports, across the population of SC neurons the mean preferred AV delay was 54 ± 30 ms (Figure 2F), which falls near the centre of the tested range (0–100 ms) and may partly reflect the distribution of delays tested.

8. Page 20, first line: Clarify if the “mean FR over a 20 ms window” refers specifically to a window centered around the peak response.

The referee is correct that we used a temporal window centred around the peak response. We have now modified the section accordingly.

The text now reads: We calculated the peak FR of such two groups by computing the mean FR rate over a 20 ms window centred around the maximum response time.

9. Citation 23 (Miller et al., 2015): The authors found “Larger multisensory responses when stronger responses were advanced relative to weaker responses.” Was this observed in the current dataset as well?

The reviewer correctly points out to previous work showing that audiovisual delay selectivity might depend on the overlapping duration of the excitatory and inhibitory periods of activity between two unisensory stimuli⁹. Additionally, it has been reported that temporal sensitivity in multisensory integration follows the ‘stronger first’ principle, where maximal multisensory enhancement occurs when stronger responses precede weaker ones¹⁰.

To investigate whether this principle applies to our data, we computed a unisensory imbalance index for each neuron, following the approach of Miller et al. (2015)¹⁰:

$$\text{Unisensory Imbalance} = \frac{V - A}{V + A}$$

We then measured the Pearson correlation between this imbalance and each neuron's preferred audiovisual delay. As shown in the figure provided, we found a significant correlation between

unisensory imbalance and preferred delay (Pearson $r = 0.104$, $p = 0.02$; slope = 0.55 ± 0.22 °/mm; offset = 3.18 ± 0.09 °), consistent with the “stronger first” effect described by Miller et al. (2015)¹⁰. However, while this trend is present in our data, we do not believe it is strong or central enough to warrant inclusion in the main manuscript. More importantly, this mechanism does not fully explain the patterns we observe. Notably, 49% of the multisensory neurons we identified exhibited no detectable unisensory responses, suggesting that classical feedforward models of multisensory integration—based solely on the relative timing and strength of unisensory inputs—are insufficient to account for the delay tuning we observe. This finding further supports the conclusion that nonlinear integration mechanisms, beyond simple temporal summation of unisensory responses, play a critical role in shaping audiovisual delay sensitivity in a substantial proportion of the multisensory neuron population.

10. Figure 2L: Add a visual indication of the mean (e.g., a small downward-pointing triangle) to the plot for clarity.

This has now been added to the figure.

11. Page 21 (CCG analysis): The authors mention, “only neurons with a baseline FR of at least 10 Hz were included.” Given supplementary Fig. 5B suggests this threshold should exclude more than half of the neurons, clarify how 3543 neurons were included.

Thank you for your comment. We appreciate the opportunity to clarify this point. The 3543 neurons included in the CCG analysis were selected based on two distinct criteria:

- Stimulus-responsive neurons (i.e., those significantly modulated by visual, auditory, or multisensory stimuli) were included regardless of their baseline firing rate, as their task-related activity indicated functional engagement in sensory processing.
- Non-responsive neurons i.e. neurons that did not have evoked responses by our stimulus, potentially including motor related neurons, were included only if they had a baseline firing rate of at least 10 Hz. This threshold was applied to reduce potential noise from very low-firing units, particularly since these neurons did not show stimulus-driven responses. Our goal was to characterize the general level of connectivity in the network while ensuring reliable signal quality.

This selection strategy applies specifically to the analyses shown in Figure 3A, 3B, and 3E, where both responsive and non-responsive neurons are included.

We have now clarified this point in the revised Methods section. However, we are happy to adjust this criterion if the reviewer believes an alternative approach would be more appropriate.

The methods now reads: *For this analysis, we included all neurons that were responsive to any stimulus (visual, auditory, or multisensory), regardless of baseline firing rate. For non-responsive neurons, only those with a baseline firing rate of at least 10 Hz were included.*

12. Page 9, Fig. 3C: Clarify the connectivity rate discrepancy—0.9% or 1.39%?

Thank you for pointing this out. The correct value is 1.39%, which refers specifically to connections between auditory-auditory (A-A) neurons. We mistakenly reported the 0.9% value, which corresponds to connections between auditory-visual (A-V) neurons. We have corrected the text to ensure clarity and consistency with the figure.

13. Page 9: "Multisensory neurons were more strongly connected with other multisensory neurons." However, Fig. 3D indicates different subtypes (D and M-D) have stronger connections to visual (V) neurons. Clarify or rephrase to state that multisensory neurons preferentially connect within their own subtype (D or M-D).

The reviewer is correct that D and M-D subtypes have strong connections to visual neurons. The previous statement pointed to the fact that together all multisensory neurons, regardless of their selectivity, receive more inputs from other multisensory neurons than from unisensory neurons (for example delay neurons receive 2.43% of inputs from D and M-D and 1.51% from auditory and visual together). However, we agree with the reviewer that the emphasis should be in the fact that multisensory neurons preferentially connect with their own subtype, and we have modified the statement accordingly. *"Interestingly, multisensory neurons preferentially connect within their own subtype and when grouping together multisensory neurons (AV delay-selective and delay-nonselective neurons) they were more strongly connected with other multisensory neurons than to unisensory neurons (Figure 3C,D)."*

14. Figure 3F: Maintain consistency in the ordering of neuron subtypes throughout figures and within individual panels for clarity.

The reviewer is right, the ordering of the neuron subtypes was swapped in Figure 3F, it has now been amended and we have also verified the rest of the figures to maintain consistency.

15. Figure 3G: Enhance the differences in line thicknesses in the schematic to make variations in connectivity clearer and easier to perceive.

We have now enhanced the differences in line thickness to improve clarity.

16. Page 15: Correct the typo "genetic market" to "genetic marker."

Apologies for the typo. This has now been corrected.

17. Page 18: Correct or update the Coliseum link (<https://github.com/FrancisCrickInstitute/Coliseum>) as it is currently nonfunctional.

The link to the stimulation apparatus has now been made public as well as our analysis codes and datasets used in this study.

Reviewer #2 (Remarks to the Author):

Bianchini et al. present the functional specialization of multisensory (audiovisual) temporal integration in the mouse superior colliculus (SC). Using electrophysiological recording with a Neuropixel probe, they recorded over 5,000 neurons across anatomical axes of the SC, demonstrating how the spatially and functionally distinct SC neurons represent the audiovisual delay. The authors provide new insights into how the SC encodes audiovisual timing, particularly emphasizing the role of nonlinear integration and regional connectivity in supporting precise temporal representation of audiovisual inputs.

More specifically, they performed analyses from single neurons to populational levels and found how the audiovisual temporal delay is differentially encoded in AP, ML, and DV locations in the SC. The data strongly support the following: 1) non-linear summation in the SC is important for decoding audiovisual temporal delay; 2) higher recurrent connectivity (more functionally connected neuronal pairs) in the medial SC; 3) in correlation with this, robust encoding of audiovisual temporal delay in the posteromedial SC.

The experimental approach is technically impressive, and the authors present a compelling case for the role of nonlinear integration and regional connectivity in supporting precise temporal representations of multisensory inputs. However, several key concerns were raised, and we hope the authors can address those by conducting additional experiments, analyzing, and clarifying or toning down interpretations to enhance the scientific clarity of the conclusion and consolidate their story.

We thank the reviewer for their insights, their supportive words and their time in assessing our work. We hope we have addressed all the concerns and in the process improved the quality of our work. We are particularly thankful for the suggestion to analyse RF similarity and putative connection probability. This has been performed in a separate dataset and shows results similar to those previously found in the visual cortex, whereby spatial receptive field correlation is a strong determinant of connection probability. The results have been now incorporated and provide further support for the analysis of putative connectivity.

1. Additional experiments

1) Causality of recurrent connectivity

While the authors observed strong correlations between encoding of audiovisual temporal delay and putative functional (synaptic) connections measured by CCG, they did not provide direct causal evidence. Furthermore, their CCG analysis is still correlative to synaptic connectivity. Additional validation (e.g., optogenetic perturbations or in vitro paired recordings) would strengthen causal claims about recurrent circuits shaping multisensory integration.

Furthermore, the authors excluded lag=0ms peaks from the CCG to avoid shared input artifacts. However, shared inputs might be segregated across SC areas and more strongly affect audiovisual integration differentially across SC regions (compared to recurrent connections). This potential must be tested or discussed further.

We thank the reviewer for the suggestion. Unfortunately, incorporating paired patch clamp recordings or optogenetic perturbations by itself would not be sufficient to demonstrate the link between functional responses and connectivity as there are presently no genetic markers from multisensory neurons. The experiments would therefore require prior functional characterisation. Imaging the SC is very challenging as it lays underneath a thick transversal sinus. Moreover, multisensory neurons are deep in the SC and not easily accessible with 2 photon imaging. Having first-hand experience performing in vivo imaging in the visual cortex followed by patch clamp recordings¹¹⁻¹³ I am as eager

as the reviewer at performing such experiments in the SC. However, I am also acutely aware from previous experience of the need for the collaboration of 2 people working full time for several years to obtain the dataset requested in a more accessible area such as the visual cortex. We have carefully referred to our estimated as putative connections in the present study and we have discussed the need for such experiments in our discussion. And we hope future work combining in vivo functional imaging and connectivity assessments will further validate our results. Our work and that of others have shown how response correlation is the best predictor of connectivity^{11,14,15} and the method used for estimating connectivity has been well characterised¹⁶⁻¹⁹. We therefore believe that the additional experiments requested are beyond the scope of the submitted manuscript.

Regarding the concern about shared inputs and our exclusion of 0 ms lag peaks from the CCG analysis, we now include results from this analysis in Supplementary Figure 6. These data suggest that shared inputs are more uniformly distributed across SC regions, in contrast to the spatially structured patterns observed for our putative local connections. Specifically, we found that the mean connection probability was similar for putatively connected pairs (2.3%) and pairs with 0 ms lag connections (1.8%). When examining how 0 ms lag connection probability varied along the mediolateral (ML) and anteroposterior (AP) axes, we found no statistically significant differences (ML: $P = 0.25$; AP: $P = 0.42$, LMM). In contrast, putatively connected pairs with non-zero lags showed a significant gradient along the ML axis ($P = 1 \times 10^{-5}$), but not the AP axis ($P = 0.2$, LMM). This pattern held when the SC was divided into three anatomical bins along each axis (ML: non-zero lag $P = 0.001$, 0 ms lag $P = 0.11$; AP: non-zero lag $P = 0.82$, 0 ms lag $P = 0.10$).

We believe these findings support the notion that shared input and recurrent connectivity contribute differently to multisensory integration across SC regions. We appreciate the reviewer's input, which helped us clarify and strengthen these points in the revised manuscript.

The revised manuscript now reads: *This spatial gradient was not observed when analysing pairs of correlated neurons with 0 ms lag, that could reflect shared inputs instead of putative connections (ML: $P = 0.25$; AP: $P = 0.42$, LMM; **Supplementary Figure 6**).*

2) No behavioral correlates

The authors proposed that the decoding accuracy of the audiovisual temporal delay in the posteromedial SC is inferred from physiology alone. To clearly support their idea, they must test behavioral correlates (e.g., reaction time or detection thresholds for AV delays) while disrupting activity correlates in the SC. The behavioral experiment can also support their ecological interpretation that visual-leading AV stimuli evoking stronger supralinear integration are linked to natural statistics of light and sound propagation.

We agree with the reviewer that behavioural correlates would strengthen our study. However, the present work provides clear evidence that the temporal difference between auditory and visual information is captured by neuronal responses in the superior colliculus, and that this is supported by the nonlinear integration of these signals at the level of individual neurons. Since our results do not indicate enhanced decoding accuracy for any specific audiovisual (AV) delay, we do not propose shorter reaction times or behavioural thresholds for particular AV delays. Rather, our hypothesis is that AV delays could bias the selection of behavioural responses. For instance, longer delays might bias responses towards escape behaviours, while shorter delays could bias responses towards freezing. Unfortunately, we encountered considerable variability in behavioural responses during tests of escape behaviour. These responses are influenced not only by stimulus properties but also by

learning history, internal state, and environmental factors, including previous habituation to the testing arena, hunger or stress levels, and distance to shelter, among many others. Importantly, any broad disruption of activity in the superior colliculus would also impact responses to unisensory stimuli, making it difficult to isolate the behavioural relevance of AV delay processing. Addressing this question would require the selective targeting of multisensory neurons in the SC. However, to date, no genetic marker specific to multisensory neurons has been identified. In a separate ongoing project, our laboratory is working to identify such a marker by combining in vivo two-photon functional imaging with in situ sequencing and precise in vivo–in vitro registration. We anticipate that this effort will take several years. If successful, this would enable the generation of transgenic mouse lines or the development of promoter-based viral targeting strategies for manipulating multisensory neurons specifically. The focus of the present study has been to demonstrate the physiological resolution with which AV delays can be discriminated by SC neurons. We are now embarking on a longer-term project aimed at investigating the behavioural relevance of these representations and how perturbations of multisensory neurons may affect multisensory-guided behaviours. At this stage, however, we believe these investigations are beyond the scope of the current study. We have now clarified this point in the discussion. As requested by the reviewer, we have also carefully moderated our interpretations to ensure scientific clarity in the conclusions.

Specifically, in the introduction we have removed the reference to orienting responses:

- 1) Our results suggest that region-specific functional specialisation in the SC underlies multisensory temporal integration, enhancing the encoding of AV information ~~to guide appropriate orienting responses.~~

The following sections have been changed in the discussion:

- 2) An accurate estimation of AV delays could serve as a mechanism for predator's distance estimation and therefore guide the rapid selection of defensive responses, such as freezing or escaping. *To validate this, future work should address the behavioural relevance of using AV delays as a cue for the selection of defensive responses.*
- 3) A highly interconnected network of multisensory neurons might provide a mechanism for extending the multisensory binding window, ensuring robust and precise encoding of sensory signals. This *could ensure* reliable detection of targets across a wide range of environmental conditions, while nonlinear integration enhances the precision of temporal information encoding.

2. Additional analysis

1) Populational receptive field topography according to SC coordinate.

The authors propose that the posteromedial SC is specialized for encoding AV delays in the peripheral sensory space. While prior work they mentioned supports this, the current manuscript lacks direct analysis linking SC coordinates to receptive field eccentricity. Spatial mapping of RF eccentricity or azimuthal bias plotted along mediolateral and anterior-posterior coordinates will strengthen the authors' claim.

We thank the reviewer for this suggestion. This was also asked by reviewer 1 and we have now incorporated Supplementary Figure 2 showing the RF location versus anatomical coordinate for a subset of experiments.

2) Contribution of supralinear vs. sublinear multisensory neurons to audiovisual delay encoding.

The identification of supralinear and sublinear multisensory neurons is an important observation. However, the manuscript does not explore how these subtypes contribute to AV delay encoding. A comparison of decoder accuracy or classifier weight between these groups, or even their distribution across regions, would enhance understanding of how nonlinear integration shapes temporal encoding.

We agree that distinguishing supralinear from sublinear neurons is important for understanding AV-delay coding. In our data, however, a given neuron's integration nonlinearity actually depends on the specific delay: as Figure 2N illustrates, at each tested delay a sizable fraction of neurons exhibits supralinear responses, while only a minority show sublinear responses. Thus, classifying cells rigidly as "supralinear" or "sublinear" across all delays would be misleading. In fact, only five neurons were sublinear ($MII < 0$) at every delay, indicating that persistent sublinearity is rare.

Because almost every neuron shifts its nonlinear profile depending on delay, we cannot form two large, stable populations to compare decoder performance fairly. Nevertheless, we do observe that overall decoding accuracy scales with the mean MII of the recorded population—populations with more strongly nonlinear (higher-MII) responses yield better discrimination of AV-delay conditions (Figure 5C).

To address the reviewer's request about spatial distribution, we have added Supplementary Figure 5: it overlays delay-specific supralinear and sublinear neurons onto the SC map. We did not observe any consistent clustering of one type versus the other at any delay, suggesting that position within the SC does not predict whether a cell will be supralinear or sublinear for a given delay.

3) Correlation between connectivity strength and receptive map overlap

The cross-correlation analysis yields putative functional connections, but it is unclear whether these connections occur preferentially between neurons with similar spatial tuning. Investigating the correlation between connection strength and the degree of receptive field overlap would provide insight into how spatial and temporal integration are jointly organized in SC microcircuits.

We thank the reviewer for this insightful comment. To assess whether putative functional connections preferentially occur between neurons with similar spatial tuning, we analysed the relationship between connection probability and spatial signal correlation, which serves as a proxy for receptive field similarity in our multisensory spatial receptive field mapping paradigm.

Using cross-correlogram analysis of spike trains recorded during the inter-trial interval, we identified putatively connected neuron pairs as previously described. We found that connected pairs exhibited significantly higher spatial signal correlation than unconnected pairs ($P = 1 \times 10^{-36}$, Kolmogorov–Smirnov test; **Figure 3C**). Moreover, the probability of a connection increased with increasing signal correlation ($P = 1 \times 10^{-4}$, permutation-based Cochran–Armitage trend test using session-shuffled null distribution; **Figure 3D**), suggesting that neurons with more similar spatial tuning are more likely to be functionally connected.

These findings indicate that spatial tuning and functional connectivity are closely aligned in SC microcircuits, supporting a model in which spatial and temporal integration are jointly organized through structured local connectivity.

4) Visual neuron proportion and decoder performance (Figure 4)

There is an apparent discrepancy between poor decoding performance by visual neurons alone (Figure 4E, G) and the positive correlation between decoder accuracy and visual neuron proportion in mixed populations (Figure 4I). A discussion or control analysis explaining how visual neurons may contribute indirectly to decoding (e.g., by stabilizing baseline population activity) would be helpful.

The reviewer is correct in this assessment. We think that visual neurons can contribute to the decoding performance when intermingled with the rest of the population. This is because when using a population decoder, the combined activity from a visual neuron and an auditory neuron will be enough to decode audio-visual delays. Our control using linearly summed responses in the decoders attempts to address this confound and demonstrates the enhanced encoding by nonlinear summation. The quantification of decoding accuracy does not specifically reflect the specific error rate between audio-visual delay and the synthetic linear sum of auditory and visual signals. Moreover, visual neurons might have some degree of multisensory modulation that when combined with auditory and delay selective neurons might enhanced the decodability of the multisensory stimulus. We have now incorporated this point into the results.

3. Clarifying methods and data interpretation

1) Decoding accuracy vs. subpopulation distribution (Figure 6C)

While the authors conclude there is no significant correlation between decoding accuracy and neuronal subpopulation distribution across SC regions, several P values, particularly for auditory neurons, are relatively low ($P=0.06$ for auditory neurons; $P=0.1$ for multisensory neurons). It would be more accurate to interpret these as trends rather than clear null results. Toning down this conclusion would avoid overstatement.

In agreement with the reviewer, we have toned down the assertion and the section now reads: *“Our correlation analysis between decoding accuracy and the distribution of subpopulations did not reach significance for any of the subpopulations, albeit suggested a trend for auditory, delay-selective and delay-non selective multisensory neurons (for percentage of visual neurons: $P = 0.4$, auditory: $P = 0.06$, delay selective: $P = 0.1$, multisensory delay-nonselctive neurons: $P = 0.1$, LMM, **Figure 6C**).”*

2) Spatially misaligned receptive fields (RFs)

All AV delay experiments were conducted using spatially coherent audiovisual stimuli, yet Figure 1H and 1L show that individual neurons' RFs are not perfectly aligned across modalities. It is unclear how such misalignment was handled in decoding and summation analysis. Did the authors exclude or control for neurons with divergent auditory and visual RFs? A more detailed description of this criterion is essential to validate the population analysis.

We included all neurons in the analysis which incorporates some variability in the spatial coincidence of the auditory and visual receptive field location. We reasoned that spatial auditory and visual RF coincidence of individual neurons might contribute to their nonlinear integration but this factor would be constant across audio-visual delays. Moreover, gated neurons, only responding to coincident auditory and visual stimuli would not have shown unisensory receptive fields. Since these neurons constitute a significant proportion of our neurons using perfect spatial coincidence as a criteria would

have not been suitable for this population. We have now clarified in the methods section this inclusion criteria for the analysis.

New wording in methods: *“Visual and auditory spatial RF overlap for individual neurons was not used as a selection criteria for the analysis of AV delay tuning.”*

3) Statistical significance in Figure 2L

The manuscript claims that observed preferred AV delays are longer than those predicted by linear summation. However, the reported mean difference (~13ms) appears modest, and no statistical test is provided. To support this conclusion, a formal statistical comparison (e.g., paired t-test or Wilcoxon signed-rank) should be included.

Thank you for the suggestion. We have now performed a Wilcoxon signed-rank test comparing the observed and predicted preferred AV delays, which confirms a significant difference ($P = 1 \times 10^{-13}$). We have updated the manuscript text to include this statistical result.

4) Decoder generalization with few neurons

The finding that only 9 neurons are sufficient to decode AV delay with above-chance accuracy is intriguing. However, more detail is needed on whether these neurons were sampled across multiple SC regions or within a single anatomical subdomain. This clarification would help interpret the generalization of local population coding.

Thank you for the suggestion. Our finding that 9 neurons are sufficient to decode AV delay with above chance accuracy was obtained by randomly sampling neurons across different SC subregions. We have clarified this in the results description. We have also included the number of neurons necessary to obtain above-chance accuracy for each of the different subregions (Supplementary Figure 9). Briefly: in the anterior-medial portion of the SC, 9 neurons are sufficient to decode AV delay; in the posterior-medial, 6 neurons; in the anterior-lateral, 17 neurons; and in the posterior-lateral, 13 neurons.

5) Minor comments

- Mismatch in data values

We found some numbers are mismatched between the manuscript and the figures.

1) [Figure 1C] The proportions of visual, auditory, and multisensory neurons do not match (13%, 9%, and 34% in the manuscript vs. 16%, 14%, and 26% in Figure 1C).

The reviewer is correct that there is a discrepancy in these values. Our previous version of the manuscript included unisensory neurons with small modulations by crossmodal stimuli in the multisensory category. However, we took a more conservative categorization in the latest version of the manuscript. None of the results change regardless of the definition of multisensory neurons. The values, and the figures are all in agreement and the codes provided reflect the work as it is presented in this revision.

2) [Figure 3C] “The percentage of connections between visual neurons was 6.17%, while auditory neurons received the highest percentage of connections from other auditory neurons, albeit their connectivity rate was much lower (0.9%)” ☒ If the latter is about the auditory-to-auditory connection, 0.9% should be 1.39%. The authors should carefully review the manuscript to ensure all numbers are correctly written.

The reviewer is correct. The correct value is 1.39% and we have now corrected the number in the manuscript. All other values in the manuscript have been proofread.

- A typo in the Discussion:

“A genetic market for identifying multisensory neurons would be a valuable tool for this purpose.”:
‘Market’ to ‘marker’

This has now been corrected.

Reviewer #3 (Remarks to the Author):

In this study, the authors propose the existence of distinct functional neuronal subtypes in the superior colliculus based on their responsiveness to different stimuli. One of these subtypes appears to be specifically tuned to asynchronous combinations of visual and auditory stimuli, enabling the encoding of temporal delays between them. Using machine learning techniques, specifically support vector machine (SVM) classifiers, the authors suggest that a subset of these neurons contains sufficient information to encode stimulus delay. This leads them to propose that the superior colliculus may employ a population coding mechanism to represent temporal delays. Their results indicate that a significant portion of the information used by the classifiers to predict delay is linked to the nonlinear summation occurring within this multisensory neuronal subtype. Moreover, they demonstrate that the capacity to encode this information is not homogeneously distributed across the superior colliculus but is particularly enhanced in its posteromedial regions. This region is associated with the upper-temporal visual field, prompting the authors to speculate that encoding delays between visual and auditory stimuli may contribute to threat detection and predator avoidance in animals.

The experiments presented in this study appear to be meticulously conducted, with a strong emphasis on data quality and methodological rigor. To my knowledge, the authors employ state-of-the-art extracellular recording technology, specifically Neuropixels probes, and utilize an automated spike-sorting method that is subsequently reviewed by human experts to exclude low-quality units. Additionally, they complement their electrophysiological recordings with video tracking of animal movements to account for potential motor influences on neuronal responses. For electrode localization, they implement an advanced approach that surpasses traditional histological techniques. A robust quality control process is also in place to exclude recordings that fail to meet predefined anatomical and physiological criteria. I commend the authors on their rigorous experimental approach.

We thank the reviewer for the positive assessment of our manuscript in particular for the appreciation of our rigorous approach. Below we provide a detailed response to the different points of concern raised.

However, I have a few concerns regarding methodological transparency and statistical analyses. Firstly, the authors do not explicitly state whether any animals or recordings were excluded from the study. While they mention performing an a priori sample size estimation, details regarding these calculations are not provided, even in the supplementary materials. Including this information would improve the study's ethical transparency and serve as a valuable example of best practices for the scientific community.

For the experiments using the AV delay protocol, a total of 106 recordings were performed on 24 animals. Of these, 14 recordings—obtained from 10 of these animals—were excluded from further analysis because the anatomical boundaries of the SC could not be determined based on visual inspection of the spike-sorted data and probe track reconstruction. The final dataset included 92 recordings from 24 animals.

With regards to the study design, we apologize for this oversight, and we have now included the relevant information in the methods section, as shown below:

We used a within-subject design, in which individual animals were exposed to all trial types (auditory, visual, multisensory, and audiovisual delays). Due to the nature of experiment monitoring during

electrophysiological recordings, the experimenter could not be blinded to condition. However, experimenters were blinded to condition during data processing, as all trial types were analysed together within each session and underwent the same processing pipeline. Importantly, data analysis was conducted blind to the anatomical location of the recording sites. Sample sizes were not predetermined but are consistent with those commonly used in the field^{1,20,21} and with current standards in mouse neuroscience studies. For analyses involving subpopulations—where neurons from different recordings were pooled—we employed a hierarchical bootstrap approach, when applicable, to account for the nested data structure (neurons within sessions, sessions within animals). In these cases, bootstrap samples were generated by first resampling animals with replacement, then resampling sessions within each animal, and finally resampling neurons within each session. Where appropriate, linear mixed-effects models (LMM) were used, incorporating recording ID and mouse ID as random effects.

Second, the study employs a clearly hierarchical dataset: neurons are nested within recordings, and recordings are nested within animals. This structure implies that neurons cannot be considered independent observations, which has important implications for statistical analyses. The authors correctly use linear mixed-effects models in some parts of the study to account for these dependencies, demonstrating awareness of the issue. However, it is unclear whether this approach is consistently applied. In particular, the use of the Kruskal-Wallis test to compare neurons raises concerns, as this test assumes independent observations. While this may not substantially alter the study's main conclusions, it increases the risk of pseudoreplication, potentially leading to inflated statistical significance and a higher likelihood of Type I errors. A thorough review of the statistical tests used would strengthen the study's methodological rigor.

We appreciate the reviewer's insightful comment regarding the hierarchical nature of our dataset. We acknowledge that neurons are nested within recordings and animals, which violates the assumption of independence in some statistical tests. We have applied linear mixed-effects models whenever the analysis was performed at single recording level, as in this case mouse and recording could be used as a random effect. For the analysis of subpopulations in which neurons from different recordings were mixed, we have now adopted a hierarchical bootstrap approach when applicable to better account for the data structure.

A third point concerns the reporting of SVM results. Confidence in the analysis would be enhanced if the authors reported both training and test accuracy, allowing readers to better assess model performance and potential overfitting. Additionally, it is unclear whether the confusion matrices presented in the figures correspond to the training or test set. Clarifying this aspect would improve the interpretability of the findings. Furthermore, while the methods section describes the dataset split (80% training, 20% testing with k-fold cross-validation) for the full dataset (5,360 neurons), it remains uncertain whether this approach is consistently applied across all SVM implementations, particularly when working with smaller subsets. The authors mention performing 20 iterations of classification in these cases, but they do not specify whether the selected neurons are randomized or remain the same across iterations. This is not a standard approach in machine learning, at least as commonly practiced, and raises concerns about its validity. Clarifying this methodological choice and providing a rationale for it would improve the study's robustness.

We thank the reviewer for their suggestion regarding reporting both training and test accuracy. To address this, we have now included an analysis (Supplementary Figure 8) showing decoding accuracy on both the training set (blue) and test set (black) as a function of the number of principal components (PCs) used in the SVM classifier, for two different neuron population sizes (25 and 50 neurons).

As expected, training accuracy remains near ceiling with increasing PCs, particularly in the larger population, reflecting the classifier's capacity to fit the training data. Importantly, test accuracy continues to increase with the number of PCs, indicating that additional components capture variance relevant to generalization rather than simply overfitting noise. This suggests that the model is still learning meaningful features from the data. Based on this finding, we chose to train the final models using all available PCs, as this consistently led to improved test performance across conditions.

We also now clarify in the manuscript that all confusion matrices correspond to the test set, and we have updated the figure legends to clearly state this.

The cross-validation approach using 80% training, 20% testing with 5-fold cross-validation was applied across all SVM implementations. Since we always repeated the stimulus 50 times, this means that 40 trials were used for training and 10 trials were used for testing in all cases.

For analyses using subsets of neurons, we fixed the number of neurons to control for decoding performance differences driven purely by population size (as shown in Figure 4). To account for variability in neuron reliability and selectivity, we repeated the classification 20 times using resampling with replacement from the relevant subpopulation. This approach allowed us to examine the effect of diverse combinations of neurons on decoding performance. We chose this method because we hypothesize that population decoding is influenced not only by the number of neurons but also by their selectivity profiles. For instance, decoding of audiovisual delays is likely improved when the sampled neurons span different delay preferences (e.g., 20, 40, and 60 ms) compared to a homogenous set (e.g., three neurons at 40 ms). Thus, our approach emphasizes the role of neuronal diversity in decoding. This has now been clarified in the methods. Moreover, we believe that this approach is common in the field, see for example Guyoton, et al., Nat. Comms. 2025²².

The number of neurons for each SVM is now reported in the corresponding figures. We have also incorporated Supplementary Figure 9 showing the SVM results for increasing number of neurons for the different subpopulations. All the results presented were qualitatively similar regardless of the number of neurons used to for the decoders as we show in Supplementary Figure 9.

Overall, the manuscript is well written and clearly structured. The introduction is well-articulated and supported by appropriate references. While multisensory integration is outside my primary area of expertise, from the perspective of a researcher interested in developmental neurobiology and circuit formation, I find the authors' effort to quantify neuronal subtype connectivity within a circuit model particularly valuable. Their findings contribute new insights into collicular circuitry in adults, providing a reference for future studies investigating superior colliculus development.

Based on an initial review of the cited literature, my impression is that the study primarily contributes to confirmatory science, reinforcing existing findings on receptive field alignment across sensory modalities, collicular neurons tuned to visual-auditory delays, nonlinear summation in multisensory neurons, and the functional specialization of different collicular regions.

We share the reviewer's enthusiasm for our findings on connectivity and the proposed circuit model. We have further strengthened this aspect of the manuscript based on the reviewers' valuable suggestions by demonstrating that connection probability increases with spatial receptive field similarity in the superior colliculus. This feature of connectivity has been previously reported in the visual cortex^{11,14,15}. Our results extend this principle to the superior colliculus and support our model of connectivity linking different sensory modalities.

We believe our confirmatory findings—including receptive field alignment across modalities, the presence of collicular neurons tuned to visual-auditory delays, nonlinear summation in multisensory neurons, and the functional specialization of collicular subregions—provide strong evidence for the robustness of our data and analyses. While consistent with previously described functional specializations across the SC, the anisotropy in multisensory integration we report—marked by enhanced AV delay decoding in specific collicular regions—has not been previously described. This highlights a novel organizational principle of multisensory processing in the SC.

Moreover, recent unpublished work²³ has questioned the presence of nonlinear integration in the mouse SC. We suggest that these discrepancies may be due to motor-related masking of auditory responses in behaving animals, and to insufficiently targeted stimulation of neurons' receptive fields. We believe our findings contribute meaningfully to this ongoing debate and provide clarity on the conditions necessary for observing nonlinear multisensory integration in the SC.

Regarding the study's conclusions, I suggest refining the strength of claims regarding population coding based on SVM classifier results. The ability of a classifier to predict stimulus delays with high precision does not imply causality or provide direct biological insight into how the information is represented in the circuit. SVM is a statistical tool designed to identify patterns in data; its success indicates that neural activity contains sufficient information to distinguish between stimuli but does not reveal the underlying computational mechanism. Multiple factors could contribute to the classifier's performance. Addressing this limitation in the discussion—along with clearer definitions of concepts such as distributed and sparse coding—would strengthen the interpretation of the results.

We acknowledge that the ability of an SVM to classify stimulus delays with high accuracy does not directly imply a specific neural coding mechanism. Rather, it suggests that sufficient information is present in neural activity to distinguish between stimuli. However, we believe that the approach does allow us to make comparisons across subpopulations and together with the functional characterisations highlights the impact of nonlinearities in the representation of sensory information and the anisotropy of this representation in the SC. We have now addressed the limitations of the approach highlighted by the reviewer in the discussion and clarified the definition of distributed and sparse coding.

This section has now been added to the discussion: *Importantly, our approach to studying population coding in the SC relied on classification methods, including SVM and RFC. While the classifiers' ability to predict AV delays indicates that neural activity contains sufficient information to distinguish between stimuli, it does not reveal the underlying computational mechanisms. Moreover, multiple factors may contribute to the classifiers' performance. To address this, we conducted analyses with a range of controls aimed at disentangling the contribution of specific factors. These analyses emphasised the role of nonlinear interactions and the distinct contributions of specific subpopulations in representing AV delays. However, the extent to which this information is utilised by downstream circuits will ultimately determine the functional relevance of this representation.*

The definition of distributed representation is included in the results section: *distributed representation, i.e. from the collective activity pattern of the population.*

With respect to the sparsity of the representation, beyond the results from the decoders, we believe this is supported by finding neurons tuned to specific AV delay. We have addressed this in the discussion: *Selectivity of AV delay tuned neurons could contribute to the sparsity of multisensory representations in the SC, by selectively increasing firing rates to specific combinations of audiovisual stimuli while otherwise maintaining a relatively low firing rate. Nonlinear selectivity can increase the dimensionality of sensory representations, enhance stimulus discriminability, minimise errors, and provide a robust neural code^{24–26}.*

In conclusion, this is a well-executed study that merits publication. However, I recommend that the authors address the concerns raised here, particularly regarding statistical analyses, methodological clarity, and the interpretation of SVM results.

We thank the reviewer for the appreciation of our study and the recommendations regarding the statistical analysis. We hope we have addressed all the concerns raised.

As a minor issue, I suggest reviewing figure labeling and references in the text. For instance, the labels for panels G, H, I, and J in Figure 2 appear to be incorrect.

We apologize for the mismatch, and we have now reviewed all the labels and references in the paper.

Reviewer #4 (Remarks to the Author):

We thank the reviewer for taking the time to review our work and we hope they found the experience stimulating and insightful.

References

1. Ito, S., Si, Y., Litke, A. M. & Feldheim, D. A. Nonlinear visuoauditory integration in the mouse superior colliculus. *PLoS Comput Biol* **17**, e1009181 (2021).
2. Ito, S., Si, Y., Feldheim, D. A. & Litke, A. M. Spectral cues are necessary to encode azimuthal auditory space in the mouse superior colliculus. *Nature Communications* **2020 11:1 11**, 1–12 (2020).
3. (PDF) Common rules of communication sound perception. https://www.researchgate.net/publication/288910900_Common_rules_of_communication_sound_perception.
4. Nadal-Nicolás, F. M. *et al.* True S-cones are concentrated in the ventral mouse retina and wired for color detection in the upper visual field. *Elife* **9**, 1–30 (2020).
5. Baden, T. *et al.* A Tale of Two Retinal Domains: Near-Optimal Sampling of Achromatic Contrasts in Natural Scenes through Asymmetric Photoreceptor Distribution. *Neuron* **80**, 1206–1217 (2013).
6. Senzai, Y., Fernandez-Ruiz, A. & Buzsáki, G. Layer-Specific Physiological Features and Interlaminar Interactions in the Primary Visual Cortex of the Mouse. *Neuron* **101**, 500-513.e5 (2019).
7. Schomburg, E. W. *et al.* Theta Phase Segregation of Input-Specific Gamma Patterns in Entorhinal-Hippocampal Networks. *Neuron* **84**, 470–485 (2014).
8. Lauer, A. M., Slee, S. J. & May, B. J. Acoustic Basis of Directional Acuity in Laboratory Mice. *JARO: Journal of the Association for Research in Otolaryngology* **12**, 633 (2011).
9. Meredith, M. A., Nemitz, J. W. & Stein, B. E. Determinants of multisensory integration in superior colliculus neurons. I. Temporal factors. *Journal of Neuroscience* **7**, 3215–3229 (1987).
10. Miller, R. L., Pluta, S. R., Stein, B. E. & Rowland, B. A. Relative Unisensory Strength and Timing Predict Their Multisensory Product. *Journal of Neuroscience* **35**, 5213–5220 (2015).
11. Cossell, L. *et al.* Functional organization of excitatory synaptic strength in primary visual cortex. *Nature* **518**, 399–403 (2015).
12. Znamenskiy, P. *et al.* Functional specificity of recurrent inhibition in visual cortex. *Neuron* **112**, 991-1000.e8 (2024).
13. Kim, M. H., Znamenskiy, P., Iacaruso, M. F. & Mrsic-Flogel, T. D. Segregated Subnetworks of Intracortical Projection Neurons in Primary Visual Cortex. *Neuron* **100**, 1313-1321.e6 (2018).
14. The MICrONS Project. <https://www.nature.com/immersive/d42859-025-00001-w/index.html>.
15. Iacaruso, M. F., Gasler, I. T. & Hofer, S. B. Synaptic organization of visual space in primary visual cortex. *Nature* **2017 547:7664 547**, 449–452 (2017).
16. Siegle, J. H. *et al.* Survey of spiking in the mouse visual system reveals functional hierarchy. *Nature* **592**, 86–92 (2021).
17. Trepka, E. B., Zhu, S., Xia, R., Chen, X. & Moore, T. Functional interactions among neurons within single columns of macaque V1. *Elife* **11**, (2022).

18. Moore, G. P., Segundo, J. P., Perkel, D. H. & Levitan, H. Statistical signs of synaptic interaction in neurons. *Biophys J* **10**, 876–900 (1970).
19. Perkel, D. H., Gerstein, G. L. & Moore, G. P. Neuronal spike trains and stochastic point processes. I. The single spike train. *Biophys J* **7**, 391–418 (1967).
20. Hoy, J. L., Bishop, H. I. & Niell, C. M. Defined Cell Types in Superior Colliculus Make Distinct Contributions to Prey Capture Behavior in the Mouse. *Current Biology* **29**, 4130-4138.e5 (2019).
21. Bimbard, C. *et al.* Behavioral origin of sound-evoked activity in mouse visual cortex. *Nature Neuroscience* **26:2**, 251–258 (2023).
22. Guyoton, M. *et al.* Cortical circuits for cross-modal generalization. *Nature Communications* **16**, 1–23 (2025).
23. Probing the integration of auditory, visual and motor signals in mouse superior colliculus. https://www.abstractsonline.com/pp8/?_gl=1*ggghpl7*_gcl_au*OTM2MDI0NzQ2LjE3MzA1NTUzMTc1MzQ1Nzc2OS4xNzMwNTUxMzMzMy*_ga_T09K3Q2WDN*MTczMDU1MTMzMjMi4xLjEuMTczMDU1MTMzMjMi42MC4wLjA.#!/10619/presentation/83309.
24. Ghosh, M., Béna, G., Bormuth, V. & Goodman, D. F. M. Nonlinear fusion is optimal for a wide class of multisensory tasks. *PLoS Comput Biol* **20**, e1012246 (2024).
25. Fusi, S., Miller, E. K. & Rigotti, M. Why neurons mix: high dimensionality for higher cognition. *Curr Opin Neurobiol* **37**, 66–74 (2016).
26. Jeffrey Johnston, W., Palmer, S. E. & Freedman, D. J. Nonlinear mixed selectivity supports reliable neural computation. *PLoS Comput Biol* **16**, e1007544 (2020).

REVIEWER COMMENTS

Below, we provide a detailed reply to the reviewers' comments. We thank the reviewers for their time and constructive feedback. We believe that addressing these comments has significantly improved the manuscript.

Reviewer #1 (Remarks to the Author):

The authors have addressed all our concerns.

We are pleased the reviewers found our corrections satisfactory and we thank them for their feedback.

Reviewer #2 (Remarks to the Author):

The authors have addressed all concerns we raised in the revised manuscript. We have only two minor comments.

1. In their response regarding the new analysis on RF similarity and connection probability, the authors mention that this was conducted in a separate dataset, stating “This has been performed in a separate dataset and shows results similar to those previously found in the visual cortex ...”. It would be helpful if the authors could clarify why the original dataset was not used for this analysis. Additionally, we recommend including details on the mice used, the number of units per recording, and the analysis performed on each group in the “Source Data” file.

We thank the reviewer for this important question and are happy to clarify. Our study comprises three complementary datasets, each designed for distinct purposes:

1. Spatial RF Dataset (979 neurons; 11 recordings in 9 animals)

This dataset was specifically designed for high-resolution RF mapping. We systematically sampled 35 spatial locations (5 elevations \times 7 azimuths), presenting auditory, visual, and multisensory stimuli with 25 repetitions per condition. This protocol lasted approximately 40 minutes per recording, allowing us to obtain robust estimates of spatial receptive field properties with high reliability.

2. Visual-leading Audiovisual Tuning Dataset (5,360 neurons; 92 recordings in 24 animals)

This dataset focused on multisensory temporal integration. We presented multiple audiovisual delays at a single spatial location to characterise AV delay tuning in detail. Prior to this protocol, we performed a coarse RF mapping (3 \times 4 locations, 4 repetitions, visual gratings drifting in multiple directions) to identify the centre of the population RF. This step was repeated as needed, and the process sometimes involved online analysis during which recording was paused; consequently, spike sorting was not applied to these brief segments. Once the population RF centre was confirmed, the AV tuning protocol began. This protocol lasted approximately **45 minutes per recording** and was optimised for probing multisensory AV delay tuning, not fine-scale RF structure (1 location, x AV delay combinations, x repetitions).

3. Visual-leading and Auditory-leading Audiovisual Tuning Dataset (1104 neurons; 22 recordings in 6 animals)

This dataset focused on comparing multisensory integration between visual-leading and auditory-leading multisensory responses (Supplementary Figure 4).

Analyses of RF similarity and its relationship to connectivity were conducted using the **spatial RF dataset**, which was designed for high-resolution RF mapping with sufficient stimulus repetitions and prolonged recording durations to ensure robust estimates. These conditions were critical for accurate RF similarity measures and for estimating connectivity via cross-correlation, which requires a large number of spikes. This approach ensures that the reported relationship is supported by the most reliable spatial RF data available.

Connectivity analyses were also performed in the **visual-leading AV tuning dataset**, but these focused on how recurrent connectivity contributes to **multisensory temporal integration** rather than RF similarity. Thus, the two datasets served complementary roles: the spatial RF dataset for spatial RF-based connectivity analysis and the AV tuning dataset for examining connectivity in relation to multisensory processing.

We have clarified this rationale in the revised Methods and Results sections.

2. In the discussion section, the authors propose that audiovisual delay may influence the decision between freezing and escaping. However, most previous studies on freezing and escape behavior have used looming shadow stimuli, whereas the current study uses a flash. Liang et al. (2015) also note that flash-evoked arrest during locomotion is behaviorally distinct from freezing. We encourage the authors to either provide additional references supporting flash-induced freezing behavior or elaborate further on how a visual flash could trigger both freezing and escape responses.

We appreciate the reviewer's point and have added references to **Storchi et al., 2020¹**, which compare responses to looming, flash stimuli, and sound. Their work shows that flash stimuli increase locomotor activity, whereas looming induces freezing, and sound produces an effect more similar to looming. In contrast Liang et al. 2015² (2015) show that a flash stimulus induces arrest. Irrespective of the effect, our argument is that the combination of visual and auditory stimuli may have different behavioural effects depending on their relative timing. Each unisensory stimulus does not necessarily need to evoke both responses. Moreover, defensive behaviours are diverse, and experimental settings often vary one stimulus parameter at a time while holding others constant (e.g., contrast, sound intensity, duration, bandwidth, tone frequency, looming, flash). Responses also vary with experience and environmental context.

We have revised the original statement:

"An accurate estimation of AV delays could serve as a mechanism for predator's distance estimation and therefore guide the rapid selection of defensive responses, such as freezing or escaping."

It now reads:

"An accurate estimation of AV delays could serve as a mechanism for predator's distance estimation and therefore modulate the expression of defensive responses, such as arrest, freezing, or escaping [Storchi et al., 2020; Evans et al., 2018; Storchi et al., 2019]."

We hope that these revisions adequately address the reviewer's comments, and we remain open to making further changes if these modifications are not considered satisfactory. We would also like to note that this statement does not alter any of the results presented in our work but rather reflects a speculative point raised in the discussion.

Reviewer #3 (Remarks to the Author):

The authors have adequately addressed the reviewers' previous comments, resulting in a significantly improved version of the manuscript. I have only identified a few very minor issues in this second review:

Page 7, first paragraph (end): The figure reference appears to be incorrect. It currently points to Supplementary Figure 2, but it likely should refer to Figure 1G.

We respectfully seek further clarification. Supplementary figure 2 is referenced in the following sections:

Overall, these results confirm the spatial alignment of visual, auditory and audiovisual RFs in the mouse SC, mapped through unisensory and multisensory stimulation. Moreover, RF location of visually and auditory responsive neurons was correlated with their anatomical location along the anterior-posterior axis of the SC (Suppl. Figure 2).

While Figure 1G shows the receptive field overlap of a visual and an auditory RF; supplementary figure 2 shows the correlation between receptive field position in visual space and anatomical location in the SC.

Otherwise, we also mention Suppl. Figure 2 here:

*Importantly, sound evoked uninstructed body movements were similar across all multisensory trial types ($P = 0.7$, Kruskal-Wallis, **Suppl. Figure 2**).*

This was incorrect and should refer to Suppl. Figure 1, and we have now changed it.

In our manuscript version Page 7, corresponds to Figure 2 and corresponding figure legend. We hope this has solved the inconsistency but we will revise further if the reviewer could clarify.

Figure 2J: The p-value displayed in the figure has not been updated to match the one reported in the main text (page 8, paragraph beginning "Given the similarity of the..."). A similar discrepancy is observed in Supplementary Figure 3, where the p-values shown in the figure do not match those in the text. This issue may affect other supplementary figures as well. I recommend a thorough consistency check. Additionally, the figure legends for Figure 2 and Supplementary Figure 3 still refer to the previous statistical method ("linear regression") instead of the updated one ("LMM").

Thank you for highlighting these inconsistencies. The p-values and statistical method labels in Figure 2J and Supplementary Figure 3 have now been corrected to match the main text and reflect the use of the updated linear mixed-effects model (LMM). We have also done a thorough consistency check across the manuscript.

Page 8, final paragraph ("Linear summation characterised..."): When reporting the MII values for conditions in which the leading stimulus is auditory or visual, the median is cited as the central tendency measure. However, no measure of variability is provided. Based on the corresponding boxplots, it appears that the interquartile range (IQR) is used and should be explicitly mentioned.

We have now modified the text accordingly:

We found that supralinear summation was weaker when auditory stimuli preceded visual stimuli (median MII auditory leading = 0.02, IQR = 0.50, median MII visual leading = 0.15, IQR = 0.41; P = 0.03, LMM; Suppl. Figure 4).

Supplementary Figure 6: The statistical test applied does not appear to have been updated to address pseudoreplication, as requested in the previous review round.

Thank you for your comment. We have now addressed the concern regarding pseudoreplication by updating the statistical analysis in Supplementary Figure 6B. Specifically, we have applied a linear mixed model, which accounts for the non-independence of repeated measurements, as requested in the previous review round.

Page 13, paragraph 2 (final sentence): The sentence “However, connectivity between delay-selective neurons was not explained by their preferred delay similarity (Suppl. Figure 5)” seems to cite the incorrect supplementary figure.

Thank you for pointing this out. We have corrected the figure citation accordingly.

Page 18, beginning of the paragraph (“Previous studies have shown asymmetries...”): Reference [46] (Bertini et al., 2008) is formatted inconsistently compared to the other citations.

Thank you for pointing this out. The formatting of reference [46] has been revised to ensure consistency with the other citations.

Reviewer #4 (Remarks to the Author):

This is a great initiative and we are glad to be part of the process.

References

1. Storchi, R. *et al.* A High-Dimensional Quantification of Mouse Defensive Behaviors Reveals Enhanced Diversity and Stimulus Specificity. *Current Biology* **30**, 4619-4630.e5 (2020).
2. Liang, F. *et al.* Sensory Cortical Control of a Visually Induced Arrest Behavior via Corticotectal Projections. *Neuron* **86**, 755–767 (2015).